



# Significant continental source of ice-nucleating particles at the tip of Chile's southernmost Patagonia region

Xianda Gong[1], Martin Radenz[1], Heike Wex[1], Patric Seifert[1], Farnoush Ataei[1], Silvia Henning[1], Holger Baars[1], Boris Barja[2], Albert Ansmann[1], and Frank Stratmann[1]

[1]Leibniz Institute for Tropospheric Research, Leipzig, Germany
[2]Laboratorio de Investigaciones Atmosféricas, Universidad de Magallanes (UMAG), Punta Arenas, Chile

**Correspondence:** Xianda Gong (gong@tropos.de)

**Abstract.** The sources and abundance of ice-nucleating particles (INPs) that initiate cloud ice formation remain understudied, especially in the Southern Hemisphere. In this study, we present INP measurement taken close to Punta Arenas, Chile, at the southernmost tip of South America from May 2019 to March 2020, during the Dynamics, Aerosol, Clouds, and Precipitation in the Pristine Environment of the Southern Ocean (DACAPO-PESO) campaign.

The highest ice nucleation temperature was observed at $-3\,^{\circ}\mathrm{C}$, and from this temperature down to $\sim-10\,^{\circ}\mathrm{C}$, a sharp increase of INP number concentration ($N_{\mathrm{INP}}$) was observed. Roughly 90% and 80% of INPs are proteinaceous-based biogenic particles at $>-10$ and $-15\,^{\circ}\mathrm{C}$, respectively. $N_{\mathrm{INP}}$ at Punta Arenas is much higher than that in the Southern Ocean, but comparable with agricultural area in Argentina and forestry environment in the US. Ice active surface site density ($n_s$) is much higher than that for marine aerosol in the Southern Ocean, but comparable to English fertile soil dust. Parameterization based on particle number concentration in the size range larger than 500 nm ($N_{>500\mathrm{nm}}$) from the global average (DeMott et al., 2010) overestimate the measured INP, but the parameterization representing biological particles from a forestry environment (Tobo et al., 2013) yields $N_{\mathrm{INP}}$ comparable to this study.

No clear seasonal variation of $N_{\mathrm{INP}}$ was observed. High precipitation is one of the most important meteorological parameters to enhance the $N_{\mathrm{INP}}$ both in cold and warm seasons. A comparison of data from in-situ and lidar measurements showed good

agreement for concentrations of large aerosol particles ($>500$ nm) when assuming continental conditions for retrieval of the lidar data, suggesting that these particles were well mixed within the planetary boundary layer. This corroborates a continental origin of these particles, consistent with the results from our INP source analysis. Overall, we suggest that high $N_{\mathrm{INP}}$ of biogenic INPs originated from terrestrial sources and were added to the marine air masses during the overflow of a maximum of roughly 150 km of land before arriving at the measurement station.

## 1  Introduction

Aerosol-cloud interaction and clouds in general still contribute the largest overall uncertainties in modelling climate change (Forster et al., 2021). A small but important subgroup of atmospheric particles are ice-nucleating particles (INPs), which can trigger cloud droplets to freeze to ice crystals above -38°C, a process referred to as heterogeneous ice nucleation. In a





mixed-phase cloud, the frozen ice crystals are supersaturated with respect to ice, and thus they grow to sizes of hundreds of
micrometers within minutes (Wegener-Bergeron-Findeisen process, Korolev, 2007). While INPs only account for very few
particles, on the order of some per cubic-meter up to some hundred per liter, depending on the freezing temperature, they can
control the initiation of precipitation and thus change the lifetime of clouds (Lohmann and Feichter, 2005).

The Southern Ocean connects all major ocean basins and is one of the drivers of the oceanic meridional overturning circula-
tion (Gent, 2016) and a major sink for carbon and heat (Frölicher et al., 2015). However, both the atmosphere and atmosphere-
ocean interactions are still poorly understood in the Southern Ocean (Xie, 2004), even though great efforts were made more
recently (Schmale et al., 2019; McFarquhar et al., 2020). The spatiotemporal distribution of aerosol particles (Yu et al., 2010;
Mehta et al., 2016) and clouds (Wylie et al., 2005) varies greatly across the Northern and Southern Hemisphere and current con-
ceptual models of cloud and precipitation formation fail to accurately reproduce the conditions in the mid and higher latitudes
of the Southern Hemisphere. Previous studies found larger ice number concentrations in mid-level stratiform mixed-phase
clouds over the Northern Hemisphere than over the Southern Hemisphere (Kanitz et al., 2011; Zhang et al., 2018; Alexander
and Protat, 2018; Radenz et al., 2021). Therefore, it is crucial to examine the population and sources of ice-nucleating particles
in the Southern Hemisphere.

A number of studies have been conducted to help us understand INPs in the Southern Hemisphere. Back in the 1970s, Bigg
(1973) measured the INP number concentration ($N_{INP}$) at $-15°C$ in the Southern Ocean (20–75 °S and 60–40 °W) and found
that it varied by 3 orders of magnitude. It was speculated that aerosol transported from distant continents is the main source
of INP. However, Schnell and Vali (1976) argued that the high $N_{INP}$ reported by Bigg (1973) was due to enhanced biological
activity, such as phytoplankton blooms. Afterwards, Bigg (1990) found a lower $N_{INP}$ compared to the previous survey (Bigg,
1973) and suggested that this was caused by changes in climate, weather systems, and transport. Recently, McCluskey et al.
(2018a) reported still lower $N_{INP}$ in the Southern Ocean than previously found and found that these INPs were mainly heat-
stable materials. Welti et al. (2020) reported on a collection of ship-based INP measurements and found that maritime $N_{INP}$
in the South Temperate Zone (23.5–66.5 °S) was lower than that in the North Temperate Zone (23.5–66.5 °N). In general,
maritime $N_{INP}$ were lower than continentally influenced $N_{INP}$, indicating that land is a stronger source of INP than the ocean
(Welti et al., 2020; Tatzelt et al., 2021). López et al. (2018) collected soil dust in northern Patagonia and confirmed its ability
to be ice active, which points out that natural mineral dust particles may be an important source of INPs in the Southern
Hemisphere.

Since the real world has large spatial heterogeneity and temporal variability, long-term measurements of INP are invaluable
and can provide a full picture of long-term INP trends and variations. For example, Wex et al. (2019) found a clear annual
variation of $N_{INP}$ in the Arctic with highest values from late spring to fall, and lowest values during winter and early spring.
The open land, as well as open water in the Arctic, were suggested as INP source regions. Welti et al. (2018) collected 4-year
samples at Cabo Verde, measured the $N_{INP}$, and found the concentration to vary up to 3 orders of magnitude at any specific
temperature. The $N_{INP}$ at any temperature followed log-normal distributions, characteristic for successive random dilution
during long-range transport. Long-term measurements are needed to understand the aerosol-cloud interaction in general and



INP in particular, and this is true worldwide and, due to a still existing lack in data, particularly in the region of the Southern Ocean.

Being located at the southern tip of South America – the only major landmass extending into the Southern Ocean – Punta Arenas is well suited to study the interaction of pristine marine conditions with terrestrial aerosol sources. In this study, ground-based and remote-sensing observations at Cerro Mirador and at Punta Arenas, respectively, were performed in the framework of the Dynamics, Aerosol, Clouds, and Precipitation in the Pristine Environment of the Southern Ocean (DACAPO-PESO) field campaign (Radenz et al., 2021). In the following, firstly we introduce the measurement sites, sampling strategy and analysis methods. Secondly, the derived INP spectra are presented, the contribution of biogenic INP is estimated and then results are compared to established parameterizations. Thirdly, processes which could possibly be controlling the presence of INP are discussed in a detailed case study for two contrasting samples, as well as for the full seasonal cycle.

## 2 Methods

### 2.1 Measurement sites

Data evaluated in this study were collected at three measurement sites. A mountain station was located at the top of Cerro Mirador (622 m a.s.l., 53.16 °S, 71.05 °W), about 8 km west of the city of Punta Arenas in southern Chile. This location at the southern tip of South America is close to the open ocean, but is nevertheless still surrounded by at least 150 km of mountainous land interspersed with fjords in all directions.

At Cerro Mirador, particle number size distributions in the size range from 10 nm to 10 $\mu$m were measured by using a TROPOS-type MPSS (mobility particle size spectrometer, Wiedensohler et al., 2012) and an APS (aerodynamic particle sizer, model 3321, TSI Inc.). Aerosols were collected on 800 nm pore size polycarbonate filters (Nuclepore Track-Etch Membrane, Whatman) with ~7-day or ~14-day time resolution and a flow rate of ~8.66 L min$^{-1}$ from 8 May 2019 to 11 March 2020. Blind filters were obtained by inserting the filters into the sampler for a period of 7 or 14 days without loading them.

Moreover, the remote-sensing supersite LACROS (Leipzig Aerosol and Cloud Remote Observations System) has been operated at an altitude close to sea level, on the campus of the University of Magallanes (53.14 °S, 70.88 °W) from November 2018 to November 2021 (Radenz et al., 2021). Measurements of meteorological parameters including temperature, wind direction, wind speed and relative humidity, cloud coverage and cloud height, were collected at the Punta Arenas Airport (53.01 °S, 70.85 °W) from SYNOP (surface synoptic observations).

### 2.2 INP measurements

After sampling, the filter samples were stored in a freezer (−20 °C). Long-term storage and transportation of the collected samples from the measurement location to Leibniz Institute for Tropospheric Research (TROPOS), Germany was always carried out in sealed plastic bags, keeping the samples frozen at all times. At TROPOS, all samples were stored at −20 °C until they were prepared for measurement.





For INP analysis, the same methods were applied as in Gong et al. (2020), namely the use of LINA (Leipzig Ice Nucleation

Array) and INDA (Ice Nucleation Droplet Array), which will both be described in more detail in the next paragraph. Concerning sample preparation, each filter was immersed into either 3 or 4 mL of ultrapure water (Type 1, Millipore) and shaken for 25 minutes to wash off the particles. 100 $\mu$L of the resulting suspension were then used for INP analysis by LINA. Further 3.1 mL of ultrapure water were added to the remaining suspension and all was then shaken again for 15 minutes. Each of 48 wells from two PCR-trays were filled with 50 $\mu$L of this suspension. Both PCR-trays were then sealed with transparent foils. One PCR-

tray was heated to 95 °C for 1 h. Both PCR-trays were then examined with INDA. Details of filter samples, including sampling number, time, and duration, total sampled volume and the air volume contributing INP to each droplet/well are summarized in Tab. S1.

The design of LINA and INDA was inspired by Budke and Koop (2015) and Conen et al. (2012), respectively. For LINA measurements, 90 droplets with the volume of 1 $\mu$L each were pipetted from the samples onto a thin hydrophobic glass slide,

with the droplets being separated from each other as each sat inside an individual compartment. The compartments were sealed at the top with another glass slide, to prevent the droplets from evaporation and ice seeding from neighboring droplets. The droplets were cooled on a Peltier element with a cooling rate of 1 K min$^{-1}$ down to −35 °C. Once the cooling process started, pictures were taken every 6 seconds by a camera corresponding to a resolution of 0.1 K. For INDA measurements, a PCR-tray was placed on a sample holder and immersed into a bath thermostat. The bath thermostat then decreased the temperature with a

cooling rate of approximately 1 K min$^{-1}$. Real-time images of the PCR-tray were recorded every 6 seconds by a CCD (charge-coupled device) camera. A LED light was fixed to the bottom of the cooling bath to cause a visible contrast between frozen and unfrozen droplets on the recorded photos. The number of frozen versus unfrozen droplets was derived automatically by an image identification program written in Python. More detailed parameters and temperature calibration of LINA and INDA, and their application can be found in previous studies (Gong et al., 2019; Wex et al., 2019). Results from all LINA and INDA

measurements are shown in Fig. S1 to S3 in the supplemental information (SI).

## 2.3 Derived INP number concentration

The possibility of the presence of multiple INPs in one vial follows the Poisson distribution. By accounting for this, the cumulative number of INP active at any temperature is obtained although only the most ice active INP (nucleating ice at the highest temperature) present in each droplet/well is observed (Vali, 1971). Therefore, the cumulative concentration of INP

($N_{\mathrm{INP}}$) per air volume or water volume as a function of temperature can be calculated by:

$$N_{\mathrm{INP}}(\theta) = \frac{-\ln(1 - f_{\mathrm{ice}}(\theta))}{V}, \tag{1}$$

with

$$f_{\mathrm{ice}}(\theta) = \frac{N(\theta)}{N_{\mathrm{total}}}, \tag{2}$$

where N$_{\mathrm{total}}$ is the number of droplets, and $N(\theta)$ is the number of frozen droplets at the temperature of $\theta$. These two parameters

are the basis for deriving temperature dependent frozen fractions ($f_{\mathrm{ice}}(\theta)$; for completeness all $f_{\mathrm{ice}}(\theta)$ spectra are included in the





SI, Fig. S1 to S3). V means the volume of air that contributed INP to each droplet/well. INDA features larger sample volumes. Assuming similar INP concentrations in each droplet or well, the larger volume used for INDA implies a higher probability of INP being present in each well, compared to each droplet examined in LINA. Consequently, INDA measurements have a lower detection limit, and are more suitable for investigating INP that are ice active at higher temperatures and are, hence, more rare.

The number of INPs present in the washing water is usually small for atmospheric samples, and the number of droplets/wells considered in our measurements is limited. Statistical errors are considered in the data evaluation. The method suggested by Agresti and Coull (1998) is used to calculate the freezing devices measurement uncertainties. Following this approach, the confidence intervals for the $f_{\text{ice}}$ can be calculated by:

$$\left(f_{\text{ice}} + \frac{z_{a/2}^2}{2n} \pm z_{a/2}\sqrt{[f_{\text{ice}}(1 - f_{\text{ice}}) + z_{a/2}^2/(4n)]/n}\right) \Big/ \left(1 + z_{a/2}^2/n\right), \tag{3}$$

where $n$ is the droplet/well number. $z_{a/2}$ is the standard score at a confidence level $a/2$, which for a 95% confidence interval is 1.96.

     For filter samples, the background freezing signal of water samples resulting from washing of blind filters is determined. Subtraction of the background was done by converting $f_{\text{ice}}$ to concentrations of INPs per volume of droplet/well, as follows (described in more detail in the SI of Wex et al., 2019):

$N_{\text{INP,corr}} = (-ln(1 - f_{\text{ice,s}}) + ln(1 - f_{\text{ice,b}}))/V, \tag{4}$

     where the corrected atmospheric INP number concentration is $N_{\text{INP,corr}}$, the frozen fractions measured for the filter samples and the field blanks are $f_{\text{ice,s}}$, and $f_{\text{ice,b}}$, respectively. In this study, all references to $N_{\text{INP}}$ refer to the corrected INP number concentrations ($N_{\text{INP,corr}}$).

## 2.4   Derived particle surface area

Particle number size distributions were obtained by inverting MPSS measurements and combining MPSS and APS measurements as described in Gong et al. (2019). These were then converted to distributions of the particle surface area concentrations by assuming particles to be spherical. Data were averaged according to filter collection times. Based on this, the ice active surface site density ($n_{\text{s}}$) (Niemand et al., 2012) can then be derived. This parameter is often used to estimate the ice nucleation activity of aerosol particles and describes the number of ice active sites per surface area, and is calculated as:

$n_{\text{s}} = \frac{N_{\text{INP}}(\theta)}{A}, \tag{5}$

where $A$ is the particle surface area concentration.

     When examining the ice activity of single types of INP in the laboratory, such as a specific mineral dust, $A$ typically is the total particle surface area for this single type of aerosol, and $n_{\text{s}}$ therefore is related to this specific material. However, for





aerosols collected on a filter in a filed campaign, $A$ originates from a mixture of several types of aerosols. In this case, $n_s$ relates
to the total surface area of both, INP and also all other particles. This has to be kept in mind when interpreting heterogeneous
ice nucleation in terms of $n_s$ in field studies.

## 2.5 Remote-sensing observations

The LACROS instrumentation comprised a PollyXT Raman polarization lidar, a CHM15kx ceilometer, a MIRA-35 cloud
radar, HATPRO microwave radiometer, and a Streamline scanning Doppler lidar. Within the present study only a few selected
products are used. The synergistic Cloudnet target classification allows for a characterization of the clouds and precipitation
above the site. Profiles of aerosol optical properties are derived from the lidar measurements with the PollyNET retrieval (Baars
et al., 2016, 2017; Yin and Baars, 2021). For this study, the particle backscatter coefficient at 532 nm wavelength was retrieved,
whenever cloud-free conditions prevailed. For the period of in-situ samples analyzed in this study, 834 of such profiles could be
retrieved. The Doppler lidar performed azimuth scans at a fixed elevation angle of $60°$ twice per hour. The azimuth-dependency
of the line-of-sight velocity is used to retrieve profiles of horizontal wind velocity and direction (Browning and Wexler, 1968).
As the signal requires sufficient amounts of particle backscatter, the maximum height of the retrieved wind profiles varies. The
lowest 500 to 1000 m height are generally covered. A comprehensive description of the instrumentation is provided in Radenz
et al. (2021).

## 3 Results and Discussion

### 3.1 Temperature spectra of $N_{\mathrm{INP}}$ and its temporal variation

The temperature spectra of $N_{\mathrm{INP}}$ (i.e., $N_{\mathrm{INP}}$ as a function of temperature) are shown in Fig. 1. All filters had INP that activated at
$-7.4\,°\mathrm{C}$, and within the detection limit, the warmest onset freezing temperature was $-3\,°\mathrm{C}$. The curves of $N_{\mathrm{INP}}$ generally span
a broad concentration range at any temperature, and below $-10\,°\mathrm{C}$ they do not intersect much. From $-5$ to roughly $-11\,°\mathrm{C}$,
some $N_{\mathrm{INP}}$ spectra increased faster than others. But generally, all spectra have temperature ranges in which $N_{\mathrm{INP}}$ increases
strongly and others in which there is a much flatter increase. Sharp increases in $N_{\mathrm{INP}}$ spectra may point to the presence of
comparably high concentrations ("comparably high" at that temperature) of one or only a few types of INP which all have
similar features causing the ice activity (e.g., the same type of ice active protein expressed by bacteria or fungi as in Knackstedt
et al., 2018). Flatter regions in $N_{\mathrm{INP}}$ spectra show up in temperature ranges in which only comparably few additional INP
become ice active with lowering temperature.

We classified samples into groups, discriminating them by concentration, depending on $N_{\mathrm{INP}}$ at $-11\,°\mathrm{C}$ to be either above
or below $5\times10^{-3}\,\mathrm{L}^{-1}$. Additionally, samples collected from May to September were assigned to the cold season and from
October to March to the warm season. Based on this, the samples were classified into four clusters, i.e., cold season with
high $N_{\mathrm{INP}}$ (CH) and low $N_{\mathrm{INP}}$ (CL), and warm season with high $N_{\mathrm{INP}}$ (WH) and low $N_{\mathrm{INP}}$ (WL), shown in different colors in
Fig. 1. During the cold season, all samples were classified as belonging to the CH cluster, except for sample 11, which had



been sampled from 14 to 22 August. During the warm season, $N_{\text{INP}}$ spectra show a broader distribution. Five samples were attributed to the WH cluster, and 13 samples to the WL cluster. It is worth noting here, that comparably high $N_{\text{INP}}$ spectra were observed throughout the year and no indication for a seasonal variation was found.

The increase rate of $N_{\text{INP}}$ (i.e., $\frac{d\log_{10} N_{\text{INP}}}{dT}$) as a function of temperature is shown in Fig. S4, compiled from all curves shown in Fig. 1. A sharp increase in this parameter can be seen from $-10$ to $-3\,°\text{C}$. At $-10\,°\text{C}$, $N_{\text{INP}}$ of most samples is already

above $10^{-3}\,\text{L}^{-1}$. For further characterizing the $N_{\text{INP}}$ spectra, the correlation coefficient of $N_{\text{INP}}$ between different temperatures is shown in Fig. S5. A strong positive correlation of $N_{\text{INP}}$ in the spectra at temperatures above $-10\,°\text{C}$ was observed. The sharp increase in $\frac{d\log_{10} N_{\text{INP}}}{dT}$ and the positive correlation at temperatures above $-10\,°\text{C}$ points towards one type of INP being present in almost all samples in comparably high amounts, which again suggests a strong and rather more local source for these INP.

Below $-10\,°\text{C}$, comparably low values and only small variations in $\frac{d\log_{10} N_{\text{INP}}}{dT}$ are seen in Fig. S4, with a minimum from

roughly $-19$ to $-13\,°\text{C}$. This temperature region can be attributed to one in which only comparably few INP from a broad range of different INP types from different sources get ice active. The curves of $N_{\text{INP}}$ below $-10\,°\text{C}$ do not intersect much, which is also reflected by a strong positive correlation of $N_{\text{INP}}$ in the spectra at temperatures below $-10\,°\text{C}$, shown in Fig. S5. However, it can also be seen in Fig. S5, that $N_{\text{INP}}$ spectra at temperatures above and below $-10\,°\text{C}$ exhibited a poor correlation with each other. This suggests that INP which are ice active in these two temperature ranges likely are of different nature and

origins.

Except for the $N_{\text{INP}}$ spectra, also distributions of all obtained $N_{\text{INP}}$ were examined separately at $-8$, $-10$, $-15$ and $-18\,°\text{C}$ (see SI, Fig. S6). Specifically, the skewness of the distribution of $log_{10}(N_{\text{INP}})$ was determined, resulting in values of $-0.9$, $-0.2$, $-0.2$ and $-0.1$ at these four temperatures, respectively. This indicates that these distributions get closer to a log-normal distribution with lowering temperatures. Log-normal distributions are expected for parameters that went through a series of

random dilutions (Ott, 1990) and they are typically observed for atmospheric distributions some distance away from sources. Therefore, INP ice active at lower temperatures probably originated from long-range transport, while a more local source can be assumed at least for the INPs active at the higher temperatures (Welti et al., 2018).

Results from previous studies on INP, added to Fig. 1, are compared with ours in the following. Tobo et al. (2013) presented primary biological aerosol particles as an important source of INP in a midlatitude ponderosa pine forest system (Manitou

Experimental Forest Observatory, MEFO) in the summertime and the measured $N_{\text{INP}}$ (cyan background in Fig. 1) in MEFO were slightly higher than ours. Note that the land cover and atmospheric environment in the MEFO site and in Punta Arenas are not comparable. Nevertheless, we here included Tobo et al. (2013) as one of their derived INP parameterizations will be discussed in Sect. 3.3. McCluskey et al. (2018a) found that the Southern Ocean has an incredibly pristine marine boundary layer with extremely low $N_{\text{INP}}$ (see magenta dots in Fig. 1). $N_{\text{INP}}$ in this study is roughly more than one order of magnitude

higher than McCluskey et al. (2018a). Welti et al. (2020) summarized ship-based INP measurements from over the world and grouped the data into different climate zones. Here we compared with $N_{\text{INP}}$ in the maritime south temperate (MST) zone and found that $N_{\text{INP}}$ in this study is generally higher than the mean value in Welti et al. (2020).

From additional atmospheric studies (not depicted in Fig. 1) it can be seen that continental aerosols generally have higher $N_{\text{INP}}$ compared to marine aerosols. For example, concentrations of INP found for Saharan desert dust aerosols (Price et al.,





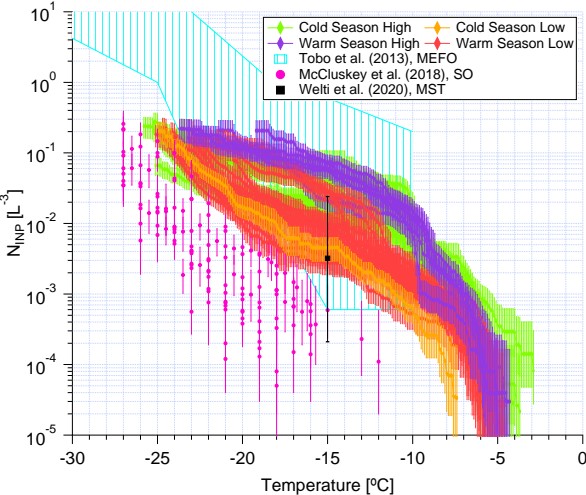

**Figure 1.** Cumulative $N_{INP}$ as a function of temperature shown as green (cold season high), orange (cold season low), purple (warm season high) and red (warm season low) lines, respectively. Error bars show 95% confidence intervals. $N_{INP}$ measured at Manitou Experimental Forest Observatory (Tobo et al., 2013), the Southern Ocean (McCluskey et al., 2018a), and in the maritime southern temperate zone (only one value at $-15\ °C$, Welti et al., 2020) are shown as cyan background, magenta dots and a black square, respectively. Error bars in McCluskey et al. (2018a) show 95% confidence intervals. Error bars in Welti et al. (2020) show 80% of the observations (excluding 10% of highest and lowest values).

2018), for biological particles from a forest (Tobo et al., 2013) or farmland (O'Sullivan et al., 2018), and for anthropogenic aerosols in the mega-city of Beijing (Chen et al., 2018) were usually higher than those from marine aerosol particles (DeMott et al., 2016; McCluskey et al., 2018a). Gong et al. (2020) examined INPs from the ocean, the ambient environment and from cloud water at the Cape Verde islands, and found that marine aerosols from sea spray production only explained a very minor fraction of $N_{INP}$ in the atmosphere and in cloud water. Tatzelt et al. (2021), having collected data in the Southern Ocean and

Welti et al. (2020), examining a broader, more global data-set found that the ship-based measurements of ambient $N_{INP}$ show 1 to 2 orders of magnitude lower mean concentrations for clean marine conditions than for continental observations. Therefore, INP in our study most likely originated from land and more evidence for this is discussed in the following sections.

### 3.2 Contribution of biogenic INPs

We examined the nature of the observed INP further. It is typically inferred that INPs that are sensitive to heat are protein-

based biogenic INPs, i.e., INPs of biological origin (Christner et al., 2008). We refer to those as bio-INP in the following. As mentioned above, in this study the particle suspensions were heated to 95 °C for 1 hour to destroy the ice activity of bio-INP. The sample was then examined in INDA to obtain $N_{INP}$ for heat-resistant INP ($N_{INP,heat\_resi}$). Figure 2 shows the time series of $N_{INP}$ for untreated samples in green dots, together with $N_{INP,heat\_resi}$ in red dots, determined at temperatures of $-8$, $-10$, $-15$ and $-18\ °C$. $N_{INP}$ varied by roughly 1 to 2 orders of magnitude at each temperature, but as discussed above, a seasonal cycle

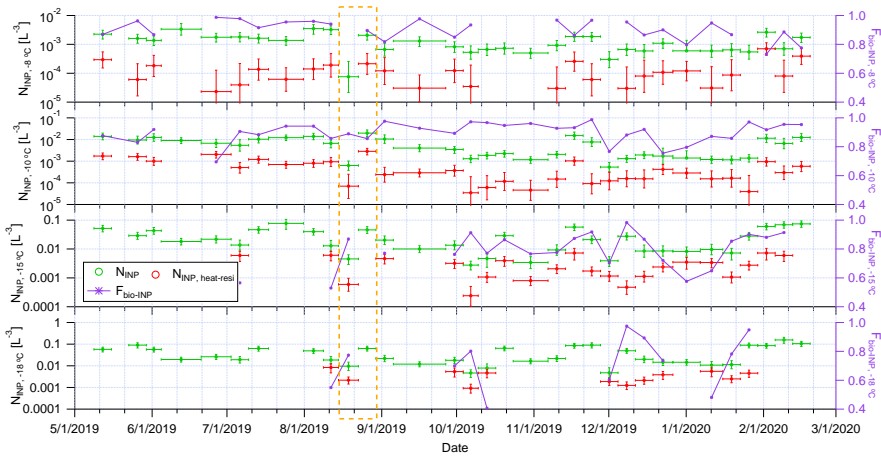

**Figure 2.** Time series of $N_{INP}$ (green circles) and heat-resistant $N_{INP}$ ($N_{INP,heat\_resi}$, red circles) at $-8, -10, -15, -18\ ^\circ C$ (from top to bottom). The fraction of bio-INP is shown as purple line. The orange box highlights samples 11 and 12, which are two samples used for a case study (discussed in Sect. 3.4).

does not become obvious. But $N_{INP,heat\_resi}$ is clearly always lower than $N_{INP}$. Therefore, Fig. 2 also shows fractions of bio-INP ($F_{bio-INP}$), calculated as:

$$F_{bio\text{-}INP} = \frac{N_{INP} - N_{INP,heat\_resi}}{N_{INP}}. \tag{6}$$

At $-8$ and $-10\ ^\circ C$, $F_{bio-INP}$ was generally above 80%. For the lower temperatures displayed in Figure 2, all wells of the PCR-trays were already frozen for a number of samples (see Fig. S3). The detection limit of INDA depends on experimental

parameters such as the droplet volume and the volume of air that contributed INP to each droplet. In this study, the upper measurement limit of INDA is roughly $10^{-2}\ L^{-1}$. Therefore, and as no LINA measurements had been done for the heated samples, values for $N_{INP,heat\_resi}$ and $F_{bio-INP}$ are not available for all samples.

In Fig. 3(a), $N_{INP}$ spectra for untreated and heated samples are shown together as black and red lines, respectively. It again becomes obvious that heating lowered the observed ice activity. Figure 3(b) shows boxplots of $F_{bio-INP}$ as a function of

temperature. The red line shows how many different samples contributed at different temperatures, and data is only shown for cases when more than half of all samples contributed. It is clear that above $-10\ ^\circ C$, more than 90% (median value) of all INP were of proteinaceous biogenic origin. From $-10\ ^\circ C$ to lower temperatures, a decreasing trend in $F_{bio-INP}$ was observed with decreasing temperature. This observation is in line with O'Sullivan et al. (2018) and Testa et al. (2021), who also found a contribution of bio-INP which was getting lower as temperatures decreased. In our study, at $-16\ ^\circ C$ generally more than

70% of all INP are still bio-INP, which is comparable to values reported by previous studies conducted in agricultural regions (Garcia et al., 2012; Suski et al., 2018; O'Sullivan et al., 2018; Testa et al., 2021).

All of this points towards a strong contribution of bio-INP to the observed aerosol, and to INP originating most likely from sources which would not be too far away and likely terrestrial. The region of the Southern Ocean is known for high fractions of





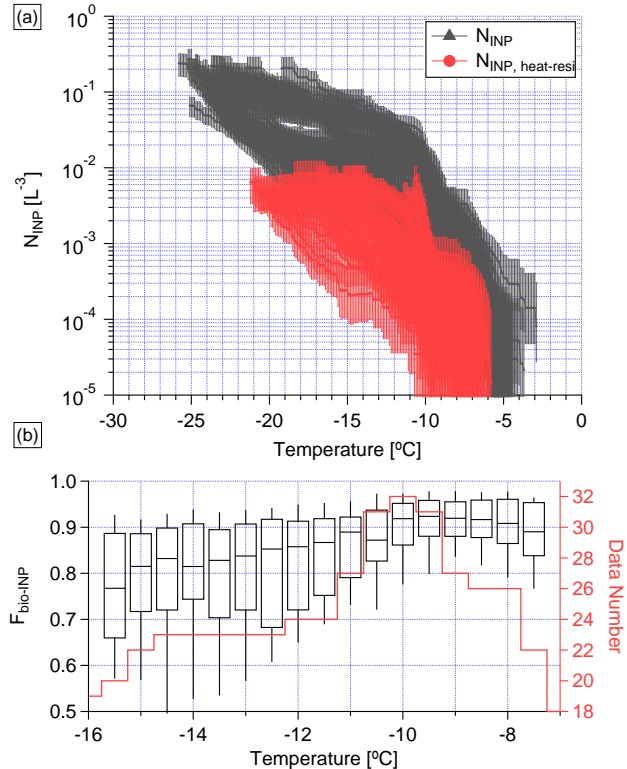

**Figure 3.** (a) Values for $N_{INP}$ and $N_{INP,heat\_resi}$ as function of temperature in black triangles and red dots, respectively. (b) Boxplot of $F_{bio-INP}$ as a function of temperature. The red line shows how many different samples contributed to the boxplot at different temperatures.

supercooled liquid clouds (Choi et al., 2010), which are thought to originate from a lack of INP connected to fewer terrestrial
sources in the Southern Hemisphere (Zhang et al., 2018). And while low $N_{INP}$ were observed in the open ocean (McCluskey
et al., 2018a; Welti et al., 2020; Tatzelt et al., 2021), we observed $N_{INP}$ that are more comparable to continental sites. When
examining satellite observations given in Choi et al. (2010) more closely, it can be seen that the effect of high fractions of
supercooled liquid clouds is less pronounced close to South America. This, together with our observations, implies that air
masses, at least those at lower altitudes, will quickly be enriched in INP when getting into contact with continental or more
generally terrestrial sources. A similar observation was also made by Tatzelt et al. (2021) based on observations performed in
the frame of a circumvention of Antarctica.

## 3.3 INP surface site density and correlation with particle number concentration

Particle number concentration for the size range above 500 nm ($N_{>500nm}$) (DeMott et al., 2010) or ice active surface site density
($n_s$) (DeMott, 1995; Connolly et al., 2009; Niemand et al., 2012) are widely used to quantify the heterogeneous ice nucleation
activity. Current models (Steinke et al., 2021) and remote sensing studies (Mamouri and Ansmann, 2015) predict $N_{INP}$ based





on either $N_{>500\text{nm}}$ or on $n_\text{s}$. Therefore, a precise understanding of $n_\text{s}$ and of the relation between $N_{>500\text{nm}}$ and $N_\text{INP}$ in different environments is highly needed.

Some previous studies found that the majority of INP were supermicron in size (Mason et al., 2016; Creamean et al., 2018; Gong et al., 2020). Therefore, we derived $n_\text{s}$ in two different ways, one using the total particle surface area and the other using

the particle surface area for particles with sizes > 500 nm. The latter surface areas were, however, roughly only about 30% lower than the former, as the majority of the particle surface area was contributed by larger particles. In Fig. 4, $n_\text{s}$ based on particle surface area for particles with sizes > 500 nm is shown, but the following discussion is generally the same for both data-sets.

Figure 4 shows $n_\text{s}$ of unheated samples as a function of temperature in black dots. Assuming the heating process only

destroys the structure of proteins but does not change the total particle surface area of an aerosol noticeably, we can also calculate $n_\text{s}$ of heated samples, shown as red dots in Fig. 4. The scatter in $n_\text{s}$ is large for both unheated and heated samples, independent of the use of the total particle surface area or only that for particles > 500 nm. But such a spread is not unusual for atmospheric data, of which we show some for a comparison to literature data in Fig. 4 as well. These literature data include a parameterization of $n_\text{s}$ for marine aerosol obtained from measurements in the Southern Ocean (McCluskey et al., 2018b). Due

to the remote location at which these data were obtained, INP likely all originated from sea spray aerosol (SSA). Also shown is a parameterization derived for English fertile soil dust (O'Sullivan et al., 2014), together with data of Saharan dust plumes from airborne measurements (Schrod et al., 2017; Price et al., 2018) and from ground-based data in marine regions influenced by Saharan dust (Cyprus and Cabo Verde, Gong et al., 2019, 2020, respectively). Although we are aware of the fact that a previous study (Testa et al., 2021) found that Argentinian soil dust contains illite and a small fraction of K-feldspar, and that these

mineral dusts may also contribute INP observed in our study, we refrain from comparing with $n_\text{s}$ parameterizations obtained from respective laboratory studies (such as Hiranuma et al. (2015) for illite NX or Harrison et al. (2019) for K-feldspar), as these relate $n_\text{s}$ purely to the surface area of these minerals, a parameter which is not available for our atmospheric data.

Coming back to Fig. 4, it is clear that at all temperatures, $n_\text{s}$ of our unheated samples is higher than that of the Southern Ocean SSA (McCluskey et al., 2018b) by up to several orders of magnitude, indicating that the aerosol we examined in southern Chile

in total is more ice active per surface area than SSA from remote regions in the Southern Ocean, i.e., the ocean surrounding southern Chile. For the English fertile soil dust (O'Sullivan et al., 2014) above about $-15\,^\circ\text{C}$, $n_\text{s}$ is roughly in the midst of our unheated data. Concerning heated data, O'Sullivan et al. (2014) found $n_\text{s}$ of heated samples to be reduced by one to two orders of magnitude, also similar to our results (data not shown as no parameterization was given). These comparisons again suggest that INP at our sampling site were influenced by terrestrial biogenic soil dust sources.

Values for $n_\text{s}$ from ground-based measurements in marine regions influenced by Saharan dust (Cyprus and Cabo Verde, Gong et al., 2019, 2020, respectively) are shown in Fig. 4, where, however, those values were adjusted such that they also relate to the surface area for particles with sizes > 500 nm. These $n_\text{s}$ values are above those for SSA, likely due to long-range transport of Saharan dust to both Cyprus and Cabo Verde. Agreement to our data can be seen for heated samples, representing the non-biogenic INP, at higher temperatures and for untreated samples only at -20 °C. At sampling sites of both Cyprus and

Cabo Verde, influence from local land masses were minimized by sampling air at the coast and upwind of the islands, different



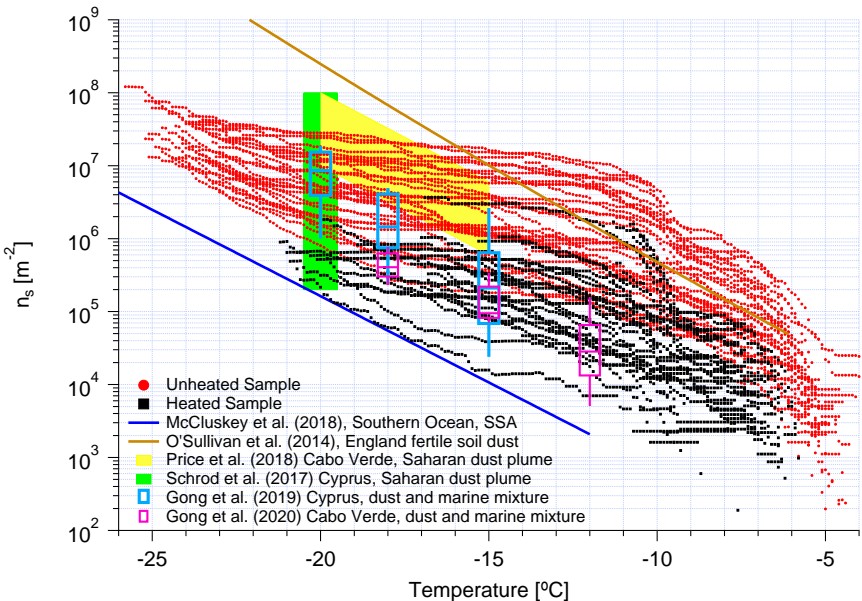

**Figure 4.** Ice active surface site density ($n_s$) of particles larger than 500 nm in diameter as a function of temperature for unheated and heated samples in black and red, respectively. Parameterizations from literature of $n_s$ for marine aerosol from the Southern Ocean (SSA) and for English fertile soil dust are shown as blue and brown lines. Ranges of $n_s$ for Saharan dust plumes are shown as green and yellow areas. Ranges of $n_s$ for ground-based data in marine regions influenced by Saharan dust are shown as blue and magenta box-plots. Respective literature is cited in the legend and in the main text. Data for Schrod et al. (2017), Gong et al. (2019), and Gong et al. (2020) are related to particle surface area > 500 nm. Data for Price et al. (2018), O'Sullivan et al. (2018), and McCluskey et al. (2018a) are related to particle surface area > 100 nm.

from the sampling site used for the here presented data. Overall, this is again one more indication that data in the here presented study were influenced by additional sources of very ice active INP.

A comparison with $n_s$ from Saharan dust plumes is only possible at $-20$ °C for Schrod et al. (2017) or below $-15$ °C for Price et al. (2018). The range of values reported by Schrod et al. (2017) covers the range in which all of our unheated

and heated data can be found. On the other hand, data from Price et al. (2018) are at the upper half of data we obtained for unheated samples, but clearly above the heated samples. Both sets of literature data are mostly much higher than our heated data, which we assumed to be representative of a non-biogenic, dust-dominated aerosol. These higher values may originate from a higher fraction of particles and total particle surface area being contributed by dust particles in the Saharan dust plumes. On the other hand, data in both Schrod et al. (2017) and Price et al. (2018) may have been influenced by biogenic contributions

to the INP, as already Kleber et al. (2007) found that protein complexes can be well preserved and possibly even accumulated when connected to mineral dust surfaces. Overall, the variability of $n_s$ for atmospheric samples, both on a local scale but also between different locations, becomes obvious by the here presented comparison.





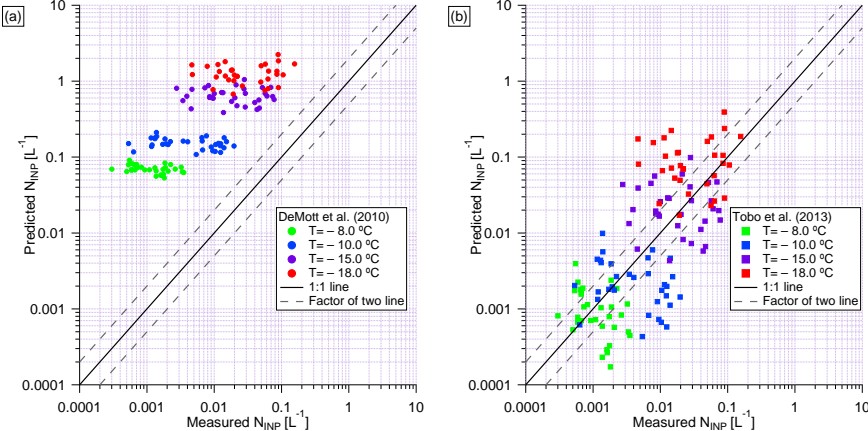

**Figure 5.** Scatter plot of measured $N_{INP}$ against predicted $N_{INP}$ obtained from parameterizations by DeMott et al. (2010) in panel (a) and Tobo et al. (2013) in panel (b).

Next, two further INP parameterizations are evaluated. Figure 5 shows the measured $N_{INP}$ against predicted $N_{INP}$ based on parameterizations by DeMott et al. (2010) and Tobo et al. (2013) in panels (a) and (b), respectively. DeMott et al. (2010)

summarized 9 field studies at different locations over the world, realized over 14 years, and proposed a fitting function to predict global average $N_{INP}$. This parameterization excluded SSA-dominated environments. It is clear that this parameterization overestimates $N_{INP}$ observed in our study by one to two orders of magnitude. Tobo et al. (2013) proposed a parameterization, based on measurements in a forest ecosystem, to predict biogenic $N_{INP}$. The predicted $N_{INP}$ is often comparable to our measured $N_{INP}$, with roughly 50% of the predicted $N_{INP}$ being within a factor of two of the measured values.

From the above comparisons between different data and parameterizations, it can be seen that there is a large variability in INP in both their concentrations and their ice activity expressed e.g., as ice active surface site density. This is rooted in the fact that many different types of particles across the range from mineral dusts to microorganisms show ice activity, and that atmospheric INP therefore originate in a multitude of different sources which have changing source strengths both in space and time. Therefore, at least the aerosol types prevailing in the area have to be considered when choosing a parameterization

to describe atmospheric INP. And in the case examined in this study, there are several indications, as discussed above, that INP in the examined aerosol, particularly those INP ice active at higher temperatures, are dominated by a strong biogenic terrestrial source present in the surrounding area.

## 3.4   Case Study

To assess the influence of meteorology and aerosol sources upon INP concentrations, we performed a case study. As mentioned

in Sect. 3.1, samples collected in the cold season have high $N_{INP}$, except for sample 11 sampled from 14 to 22 August 2019. Here we compare several meteorological and aerosol-related parameters for sample 11 and sample 12, the latter sampled from 22 to 29 August 2019. Figure 6 (a) shows the probability density functions (PDFs) of temperature, cloud height, cloud cover,





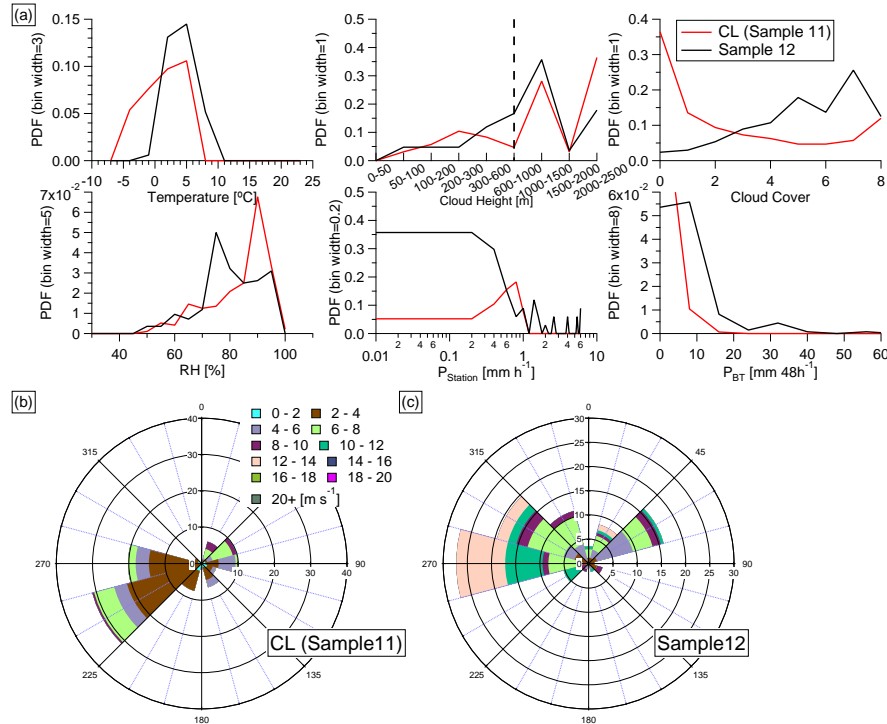

**Figure 6.** (a) Probability density function (PDF) of temperature, cloud height, cloud cover, RH, precipitation measured at the station ($P_{\text{Station}}$), and cumulative precipitation along the backward trajectory in the past 2 days ($P_{\text{BT}}$) during samples 11 (red) and 12 (black). The dashed line in the cloud height panel shows the measurement station height. (b) The wind rose plot during sample 11. (c) The wind rose plot during sample 12.

relative humidity (RH), precipitation in the station ($P_{\text{Station}}$), and the cumulative precipitation along the backward trajectory in the past 2 days ($P_{\text{BT}}$) during sample 11 (in red) and sample 12 (in black).

Surface temperatures, which were taken from a station at the Punta Arenas airport, were slightly lower during the period of sample 11 (Fig. 6 (a)). In terms of cloud cover, given as octa values following Jones (1992), the station observations (see Fig. 6 (a)) and the remote-sensing observations (see Fig. 7 (a, d)) both show a significantly higher cloud cover during the sampling period of sample 12. Especially, a long period with a cloud-free boundary layer during sample 11 is a striking feature, which coincides with high surface pressure values, indicative for calm weather conditions.

Precipitation was more frequent and more intense during sample 12 compared to the week before, which can be seen from PDFs of $P_{\text{Station}}$ and $P_{\text{BT}}$ in Fig. 6 (a), and time series of precipitation derived from the Cloudnet target classification in Fig. 7 (b, e). When rain occurs, particles can be washed out due to impaction, turbulent and Brownian diffusion as well as phoretic phenomena or simply because they acted as CCN during cloud formation. These processes depend on the size, chemical composition, and concentration of particles (Pruppacher and Klett, 2010; Chate et al., 2011). However, raindrops can disperse
soil bacteria and other bio-aerosol into ambient air (Joung et al., 2017). Furthermore, rain impaction on plants might contribute





to aerosolization of biogenic INP (Huffman et al., 2013; Prenni et al., 2013; Tobo et al., 2013; Testa et al., 2021). RH is also known as a factor affecting the INP population in a number of ways (Huffman et al., 2013; Testa et al., 2021). Testa et al. (2021) found that $N_{\text{INP}}$ at $-12$ °C is positively correlated with RH, but $N_{\text{INP}}$ at lower temperatures ($<-20$ °C) is negatively correlated with RH. In this study, we found RH is somewhat higher for sample 11 while $N_{\text{INP}}$ at $>-20$ °C is lower than in
sample 12.

During the collection time of sample 11, low wind speeds $< 10$ m s$^{-1}$ prevailed and the wind most often came from southwest (Fig. 6 (b) and Fig. 7 (c)). Boundary-layer wind shear, as it can be inferred from the difference of the wind values at the three different height levels, was also low, indicating only weakly turbulent conditions. The frequency of backward trajectories in sample 11 (Fig. 8 left panel) is in good agreement with the histogram of wind directions. In sample 12, the
wind direction mostly was from the west with a stronger wind speed (Fig. 6 (c) and Fig. 7 (f)). Compared to sample 11, the wind shear is higher as well (Fig. 7 (f)). The frequency of backward trajectories in sample 12 (Fig. 8 right panel) shows that air parcels traveled from the west to the measurement station. Considering the local geography, air parcels collected for both samples covered roughly the same distance over mountainous land, interspersed with fjords, in advance to arrival at the sampling site.

The measurement station was exposed to free-tropospheric conditions for different amounts of time during the two examined weeks. Typically, the boundary layer top is associated with a notable decrease of particle backscatter (Baars et al., 2008). As becomes evident from Fig. 7 (a), for sample 11 boundary layer heights showed a strong diurnal cycle with daytime maxima around 1100 m height and nighttime minima of 500 m or below. Thus, the top of Cerro Mirador was influenced by the free troposphere for a significant fraction of the sampling period. During sample 12, the boundary layer tops were significantly
higher, peaking at 1900 m during daytime, while nighttime lows were rarely below 700 m, leading to the sampling of mostly boundary layer air.

Figure 9 shows the PSSD (particle surface area size distribution) and the vertical profiles of particle backscatter coefficients for samples 11 (red line) and 12 (black line). PSSDs are bimodal with a minimum around 300 nm. The concentration of the larger mode is slightly higher in sample 12 than in sample 11. Also, the particle backscatter coefficient is higher throughout
the boundary layer depth and above, indicating a generally higher aerosol load observed for sample 12.

In summary, the examination of meteorological conditions and aerosol features of the two subsequent samples from the cold season with strongly contrasting $N_{\text{INP}}$, low for sample 11, high or sample 12, yielded differences. For sample 12, simultaneously to high $N_{\text{INP}}$ higher wind speeds prevailed, precipitation was more frequent and concentrations of large particles ($> 500$ nm) were slightly enhanced. A general enhancement of aerosol concentrations is evident throughout the boundary layer, as indicated
by higher particle backscatter coefficients. As discussed above, it is known that rain events can lead to more bio-aerosols, including soil bacteria, being ejected into ambient air, some of which may be INPs. In summary, higher $N_{\text{INP}}$ may have been caused by the addition of soil dust or biogenic particles over land to the sampled air masses. However, as the comparison of only two samples is of limited value, an additional discussion concerning processes which may control $N_{\text{INP}}$ are given in the following section.

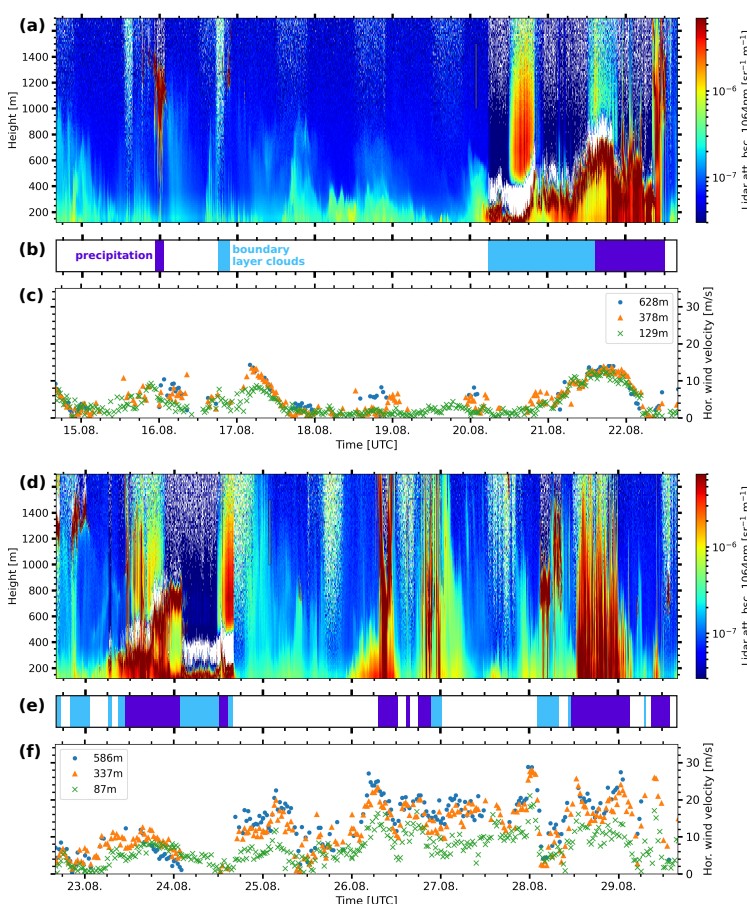

**Figure 7.** Overview of aerosol and cloud conditions at Punta Arenas during which sample 11 and 12 were collected: (a, d) Time-height cross-section of attenuated backscatter at 1064 nm at 30 s temporal and 14.9 m vertical resolution obtained from the ceilometer. (b, e) periods of boundary layer clouds and precipitation derived from the Cloudnet target classification. (c, f) horizontal wind velocity retrieved from the Doppler lidar scans at heights of 129, 378, and 628 m.





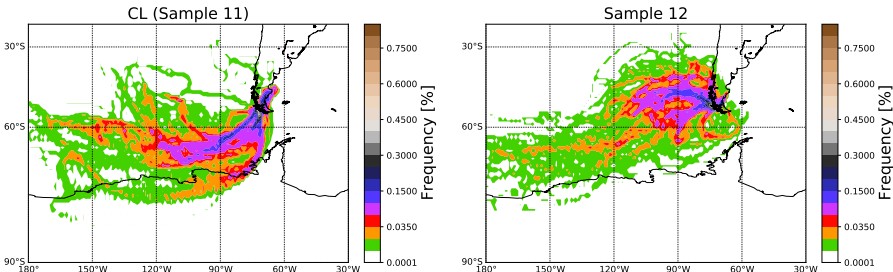

**Figure 8.** Relative frequency of backward trajectories ending at 600 m with 1 h resolution, based on a 1° by 1° grid size during sample 11 (left) and 12 (right).

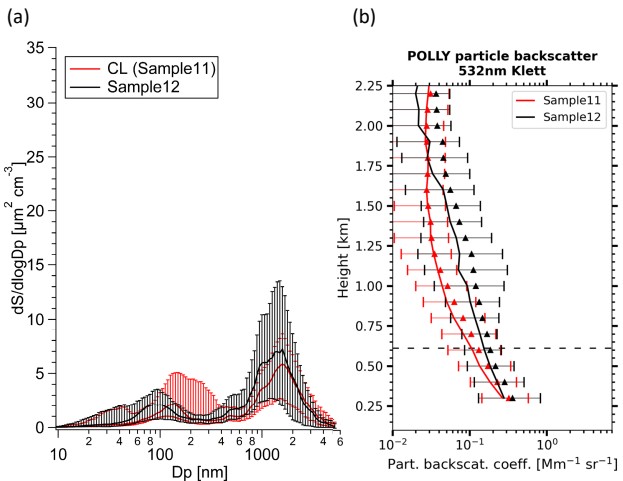

**Figure 9.** (a) Particle surface area size distribution for samples 11 (red) and 12 (black). (b) Vertical profile of particle backscatter coefficient during samples 11 (red) and 12 (black). The triangles show the mean values, and the dashed line shows the height of the Cerro Mirador station. The solid lines in both panels show the median value and error bars show the 25th to 75th percentile.

## 3.5 Processes controlling $N_{INP}$

To obtain additional insights into processes controlling the INP population, we studied the parameters that were examined for the case study in the previous section also for all samples. For that, samples were classified into four clusters, i.e., WH, WL, CH, and CL, based on $N_{INP}$, as described in Sect. 3.1.

As data for the only case attributed to CL was already shown in the previous section, Figures 10, 11 and 12 show averaged results for WH, WL and CH, only. Some generally expected differences between warm and cold seasons become obvious. Both warm season clusters had similar higher temperatures and similar lower RH than both cold season clusters (Fig. 10 (a)). Similarly, both warm season clusters had similar bi-modal PSSD, with particle number and surface area concentrations being ∼2 times higher than for both winter clusters (Fig. 12 (a)). They also had similar but higher values of particle backscatter



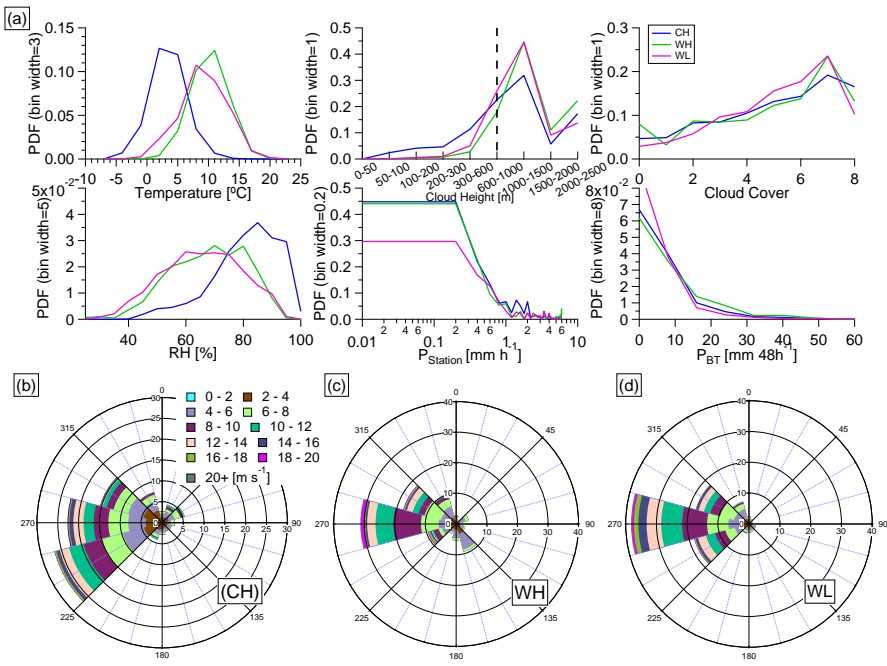

**Figure 10.** (a) Probability density function (PDF) of temperature, RH, cloud cover, cloud height, precipitation measured at the station ($P_{Station}$), and cumulative precipitation along the backward trajectory in the past 2 days ($P_{BT}$) averaged for all samples collected during cold season high (CH, blue lines), warm season high (WH, green lines), and warm season low (WL, magenta lines). (b, c, d) The wind rose plots for CH, WH and WL.

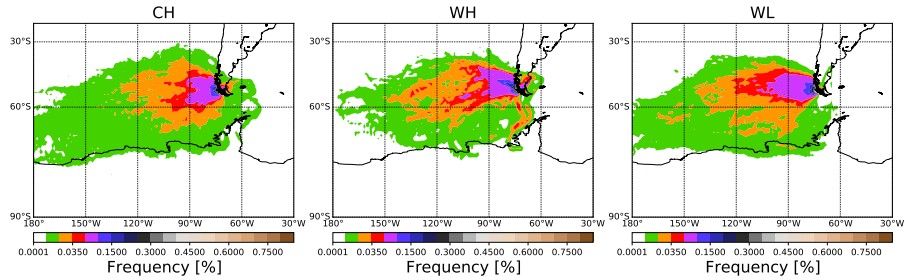

**Figure 11.** Relative frequency of backward trajectories ending at 600 m with 1 h resolution, based on a $1°$ by $1°$ grid size during cold season high (CH, left), warm season high (WH, middle), and warm season low (WL, right).

coefficients at all heights, visible in the vertical profiles shown in Fig. 12 (b). The similarity in particle abundance for both WH

and WL shows that parameters related to particle abundance alone likely cannot predict INP concentrations well for the here presented data.





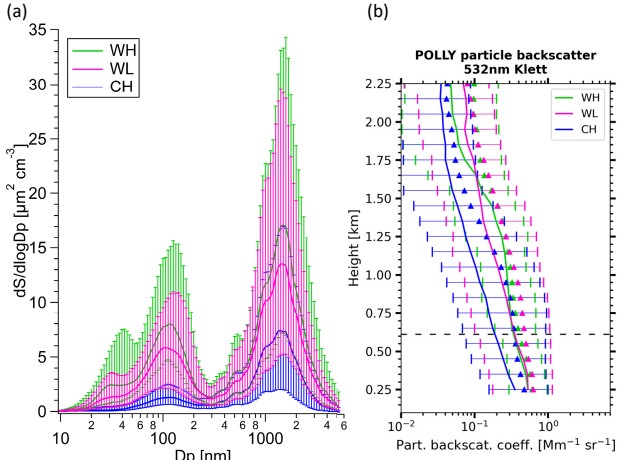

**Figure 12.** (a) Particle surface area size distribution for cold season high (CH, blue), warm season high (WH, green), and warm season low (WL, magenta). (b) Vertical profile of particle backscatter coefficient during CH, WH, and WL. The triangles show the mean values, and the dashed lines show the height of the Cerro Mirador station. The solid lines in both panels show the median value and error bars show the 25th to 75th percentile.

Similar cloud heights were observed for all clusters, with clouds at roughly 600 to 1500 m. The overall cloud cover during WH, WL and CH was comparable (Fig. 10 (a)) but higher than during CL (Fig. 6 (a)). Similarly, the wind rose plots show that wind speeds during WH, WL, and CH were comparable (Fig. 10 (b)), but higher than during CL (Fig. 6 (b)). While 37%
of the time the observed wind speeds during CL were below 4 m s$^{-1}$, less than 15% were so low during CH. During the warm season, such low wind speeds were never observed. The wind direction during WH, WL, and CH were mainly from the west, but wind directions during CL were mainly from the southwest. As for the case study, also for all clusters the backward trajectory frequencies shown in Fig. 11 are in good agreement with wind directions shown in the wind rose plots. During WH, WL, and CH, the air parcels featured similar paths, mainly originating from the Southern Ocean to the west of the sampling
location on Cerro Mirador. Also, the vast majority of trajectories stayed north of 65 °S for these three clusters, while more than 20% of trajectories during CL passed much further south. None of these parameters related to clouds or wind speed or direction gives clear insights into why WH and WL featured so clearly different $N_{\mathrm{INP}}$.

Based on lidar data, further agreement for WH, WL and CH was seen. With exception of sample 11, no extended periods of boundary layer top heights below 800 m were observed. Also, mineral dust in the free troposphere was not observed during
the sampling period (Radenz et al., 2021). This and the above discussed origin of the backward trajectories makes it unlikely that observed high concentrations of INP originated from mineral dust particles.

Finally, $P_{\mathrm{Station}}$ and $P_{\mathrm{BT}}$ during WH and CH were higher than during WL and CL. Of all discussed parameters, precipitation was the only one that distinguished high from low $N_{\mathrm{INP}}$. As mentioned before, raindrops can disperse soil bacteria and other bio-aerosol into ambient air, and also rain impaction on plants might contribute to biological INP. We suggest that precipitation
indeed was an important meteorological parameter affecting INP concentrations in this study.



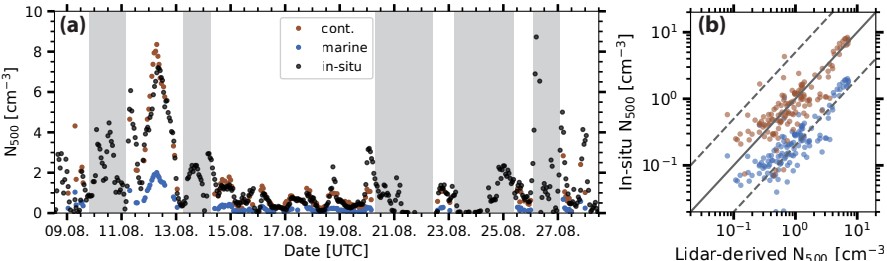

**Figure 13.** (a) Time series (during sample 10, 11, and 12) of in-situ measured $N_{>500nm}$ in black dots, and lidar retrived $N_{>500nm}$ when assuming a lidar-ratio typical for marine or continental particles in blue and brown dots, respectively. (b) The in-situ measured $N_{>500nm}$ against the lidar retrived $N_{>500nm}$ when assuming a lidar-ratio typical for marine or continental particles (during sample 10, 11, and 12) in blue and brown dots, respectively.

To sum up, we found that the meteorological parameters, including cloud height, cloud cover, and RH are not directly related to $N_{INP}$. Instead, we suggest that strong precipitation is one of the most important parameters to enhance the $N_{INP}$ both in cold and warm seasons. The air masses are mainly coming from the Southern Ocean, without difference during cold and warm seasons. The particle surface area concentration in the cold season is much lower than in the warm season. However, the $N_{INP}$ during the cold season are all in the higher concentration cluster, except for one sample.

### 3.6  Relationship between in-situ measured $N_{>500nm}$ and lidar data

In this section, $N_{>500nm}$ obtained from the in-situ measurements is compared with values obtained from lidar data. The retrieval of the latter is based on the profiles of particle backscatter coefficients at 532 nm (Sect. 2.5). The extinction coefficient is then calculated using typical extinction to backscatter (lidar-) ratios for marine (20 sr) and continental (50 sr) aerosols (Müller et al., 2007; Bohlmann et al., 2018). In our study, the optical properties were then taken for a height of 622 m above the remote sensing site. The optical properties were converted to particle number concentrations with the conversion factors for marine and continental aerosol, respectively, as given by Mamouri and Ansmann (2016).

Figure 13 (a) shows an exemplary time series (during sample 10, 11, and 12) of in-situ measured (black dots) and lidar-retrieves (a lidar-ratio typical for marine particles in blue dots and for continental particles in brown dots) $N_{>500nm}$ and (b) shows the corresponding scatter plot. It can be seen that values retrieved when assuming continental aerosol fit well to the in-situ data, while assuming marine aerosol resulted in clearly too low $N_{>500nm}$. Figure 14 shows the in-situ measured $N_{>500nm}$ against the lidar retrieved $N_{>500nm}$ when assuming a lidar-ratio typical for marine or continental particles in blue and brown dots, respectively, including all data available from May 2019 to March 2020. A linear fit in the logarithmic space through the continental data shown in Fig. 14 yields an R-value of 0.72 with a slope of 0.99 and an intercept of 0.003. Therefore, we find that $N_{>500nm}$ retrieved from lidar data when assuming a lidar-ratio typical for continental aerosol generally fits well to the co-located in-situ data. As this retrieval was done for an altitude of 622 m, i.e., well elevated above the lidar location, it can be concluded from the comparison that the boundary layer was typically well mixed concerning large aerosol particles of



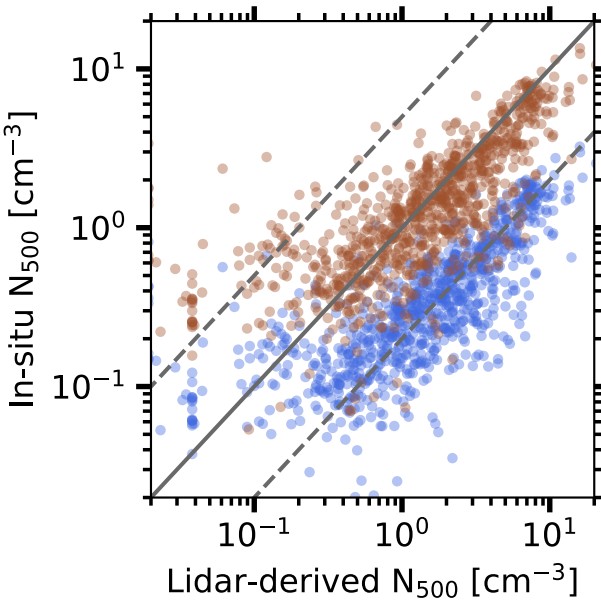

**Figure 14.** In-situ measured $N_{>500nm}$ against the lidar retrieval $N_{>500nm}$ when assuming a lidar-ratio typical for marine or continental particles (including data available from May 2019 to March 2020) in blue and brown dots, respectively.

continental origin. Generally, indeed, a well-mixed boundary layer also was seen during most of the observation periods, as the particle backscatter coefficients were often constant for altitudes between 250 m and 1.2 km (as also indicated in Fig. 3 of Radenz et al., 2021).

The lidar comparison suggested that the large aerosol particles (>500 nm) are of continental origin, which is in consistency with the INP source analysis. Moreover, INP concentrations could generally be derived well based on in-situ derived $N_{>500nm}$ values when using the parameterization from Tobo et al. (2013) (as discussed in Sect. 3.2). Together, these results suggest that, for this study in the region of Punta Arenas, $N_{INP}$ valid throughout the boundary layer can generally be derived based on lidar

data when assuming the presence of continental aerosol in the retrieval of $N_{>500nm}$.

## 4    Conclusions and outlook

This study analyzed annual off-line measurements of INP number concentration ($N_{INP}$) that were performed in the framework of the Dynamics, Aerosol, Clouds, and Precipitation in the Pristine Environment of the Southern Ocean (DACAPO-PESO) campaign at Cerro Mirador near Punta Arenas. Measurements of aerosol particle surface area size distribution (PSSD), syn-

ergistic radar and lidar observations, backward trajectory calculations, and surface observations of meteorology (temperature, RH, cloud cover, cloud height, precipitation) were used to put obtained $N_{INP}$ into context.





The highest ice nucleation temperature we observed was $-3\,°C$ and all samples activated at $-7.4\,°C$, where it should be mentioned that these values are instrument dependent. The INP spectra ($N_{\mathrm{INP}}$ as a function of temperature) showed a sharp increase of $N_{\mathrm{INP}}$ from this measurement onset down to $\sim -10\,°C$. $N_{\mathrm{INP}}$ in this study are much higher than that in the Southern Ocean (McCluskey et al., 2018a), but comparable with measurements in an agricultural area in Argentina (Testa et al., 2021) and a forestry environment in the US (Tobo et al., 2013). No clear seasonal variation was observed. At high temperatures ($>-10\,°C$), roughly 90% of all INP were proteinaceous and hence of biogenic origin. This fraction lowered to $\sim70\,\%$ between $-10\,°$ and $-16\,°C$, indicating an important contribution of biogenic INP all over this temperature range.

Ice active surface site density ($n_{\mathrm{s}}$) in this study is much higher than for clean marine aerosol in the Southern Ocean, but similar to English fertile soil dust samples. Similar to observations reported for the latter, $n_{\mathrm{s}}$ for heated samples also decreased by one to two orders of magnitude, induced by the destruction of proteinaceous INP by the heat treatment. Values for $n_{\mathrm{s}}$ for heat treated samples are also closer to those for marine aerosols that are slightly influenced by mineral dust particles. The parameterization based on particle number concentrations in the size range larger than 500 nm ($N_{>500\mathrm{nm}}$) from DeMott et al. (2010), representing global $N_{\mathrm{INP}}$ in non-marine environments, overestimated measured $N_{\mathrm{INP}}$ by up to two orders of magnitude. However, the parameterization from Tobo et al. (2013), representing biological particles from a forestry environment, yields $N_{\mathrm{INP}}$ comparable to values obtained in this study.

Backward trajectories showed that air masses generally came from the Southern Ocean, mainly from west or southwest directions, passing at least 150 km of mountainous land, interspersed with fjords, between the ocean and the measurement site. Comparison between comparably high and low $N_{\mathrm{INP}}$ in both the cold and warm season pointed towards precipitation as the sole main meteorological parameter which may explain the observed variations in $N_{\mathrm{INP}}$. Mobilization of INP due to precipitation has been described before, and overall we suggest that high $N_{\mathrm{INP}}$ of biogenic INP originated from terrestrial sources and were added to the air masses during the overflow of land, previous to arriving at the measurement site. When assuming that aerosol particles are of continental origin, the lidar-derived $N_{>500\mathrm{nm}}$ values fit well to the in-situ $N_{>500\mathrm{nm}}$. This comparison also suggests that particles $>500$ nm were well mixed in the planetary boundary layer. We conclude that for the here in presented dataset, $N_{\mathrm{INP}}$ in the planetary boundary layer could be derived based on lidar retrieved $N_{>500\mathrm{nm}}$ and the INP parameterization in Tobo et al. (2013).

This study shows a significant land source of INP at the tip of southern South America. Using a combination of meteorological data, backward trajectories and lidar analysis, this study provides insights into the mechanisms controlling the observed INP. However, considering the complex nature and diverse aerosol sources, open questions remain. One interesting detail from our study is that the cold season shows generally lower particle number concentrations, but almost always high INP concentrations. Atmospheric INP concentrations, sources of INP and processes controlling these parameters cannot solely be answered by one study even if the study covered an annual data set. Further studies are needed, also in this region. Such studies should include a chemical analysis of the aerosol to maybe help to better understand INP sources. Also, a higher time-resolution of INP measurements, together with additional parameters such as those used here, could help to gain more detailed insight into nature and sources of INP.





*Data availability.*  INP and meteorology data are available through the World Data Center PANGAEA (https://www.pangaea.de/). A link to the data can be found under this paper's assets tab on ACP's journal website. The Cloudnet datasets are provided by the ACTRIS Data Centre node for cloud profiling via the following links: https://hdl.handle.net/21.12132/2.b6c194d7d33b448e (last access: 27 January 2022). In the upcoming future, the Doppler lidar and PollyNET datasets will also be available via ACTRIS. Meanwhile, they can be obtained upon request
from polly@tropos.de.

*Author contributions.*  X. Gong wrote the manuscript with contributions from H. Wex, M. Radenz, P. Seifert and F. Stratmann. X. Gong performed the INP measurements. F. Stratmann and S. Henning performed the SMPS and APS measurements in Punta Arenas. B. Barja collected the filter samples in Punta Arenas. P. Seifert, M. Randenz, B. Holger, and A. Ansmann analyzed the remote sensing data. F. Ataei collected the meteorology data. All co-authors proofread and commented on the article.

*Competing interests.*  The authors declare that they have no conflict of interest.

*Acknowledgements.*  We would like to thank Raul Perez, Isreal Villa, and Gonzalo Mansilla form Laboratorio de Investigaciones Atmosféri-cas, Universidad de Magallanes (UMAG), Punta Arenas, Chile for collecting filter samples. We would like to thank Thomas Conrath from TROPOS to help us maintain the SMPS, APS, CCNC and other instruments in Punta Arenas. B. Barja acknowledges partial support from ANID/FONDECYT through grant 11181335. We acknowledge the provision of data and scientific support from the BMBF-funded project
CLOUD 16 (01LK1601B) and the EU FP7 ITN-project CLOUD-MOTION (764991).



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
