# Peer review of "Significant continental source of ice-nucleating particles at the tip of Chile's southernmost Patagonia region"

_Atmospheric Chemistry and Physics, 2022_

## Author Comment (AC1)

Dear Reviewer,

We thank you for doing this review and for your suggestions that helped to improve our manuscript. Below, please find your original comments in blue and our responses in black. When referencing page and line numbers, we are always referring to the old version of the manuscript.

In this manuscript, the ambient ice-nucleation particle (INP) concentration and ice nucleation efficiency of aerosol particles collected from the southern tip of South America are presented along with some complementary field observations from three different locations in the studied region. Key findings out of this study are (including but not limited to):

*A majority of assessed INPs show the indication of proteinaceous component (evidenced by the freezing assay with heat treatment), and

*the field observations imply that the mixing of large aerosol particles from the continental source with other sources occur in the planetary boundary layer over the study region, and the authors link the observations to their INP data implications (which are detailed in the manuscript).

These findings are relevant to the ACP journal scope as ACP supports many INP-related papers. The INP data from the studied region, especially relatively long-term data as presented in this manuscript, is rare and invaluable. While the field data and offline lab measurements are reasonably well-blended, the manuscript is a bit lengthy, and this reviewer identified some speculative parts (discussed below). For that matter, this manuscript may fit more as the 'measurement reports' manuscript type under ACP, in this reviewer's opinion. While this reviewer does not have a strong opinion to turn this research article into a different manuscript type, the authors may consider it if the editor and other reviewers address a similar point. Nevertheless, the manuscript is overall well-written, and this reviewer can support the publication of this manuscript (in any manuscript type) in ACP after minor and technical revisions.

We prefer to keep the manuscript as a research article. We think this manuscript reported substantial new results of ice-nucleating particles in the southernmost Patagonia regions, where

long-term measurements are scarce. We present that continental biological aerosol, driven by raindrops, is the main source of INPs in this region, as the reviewer summarized. Additionally, in-situ data were combined with lidar data to examine the extent to which particle (and with this INP) concentrations can be derived from lidar data. This research article includes substantial advances and general implications for the scientific understanding of INP sources and properties in the Southern Ocean. It fits the scope of research articles.

**[Minor comments]**

P1L22-23: Pore condensation freezing can occur below -38 dC as long as the ice saturation condition is met. For that matter, the authors' statement, which sounds limiting the heterogeneous ice nucleation (and/or cloud droplet freezing) to the temperatures above -38 dC seems misleading. The reviewer suggests to the authors to rephrase this sentence to focus on discussing only immersion or clarify it differently.

Thanks for the clarification. We removed "above -38 °C" in the text.

P2L51: The reviewer agrees with the authors about the importance of long-term INP measurements/monitoring. The authors might want to discuss the importance of highly time-resolved INP data somewhere in this manuscript as well. In the reviewer's opinion, both long time coverage and fine time resolution of INP data are currently missing in the atmospheric science community. The reviewer thinks the time resolution employed for aerosol sampling in this study (P3L77) is valid and the associated offline INP data generated out of a week or two aerosol samples are still invaluable. Some extra discussion regarding the time resolution would add more value to the current manuscript, and the readers will appreciate seeing the discussion of limitations and future outlook.

Thanks for your suggestions. We modified this paragraph and added the following in P2L57.

"Many long-term measurements in the past years were done with off-line techniques (such as Schrod et al., 2020; Schneider et al., 2021; Testa et al., 2021), offering a comparably low time

resolution. Nevertheless, highly time-resolved data are also important. However, high time resolution INP measurements are challenging, because they often require the presence of an operator during the measurement. Also, they can only detect comparably high INP concentrations, due to optical detection, and therefore often only yield data at low temperatures or in environments where concentrations are high. Recently, newly developed INP instruments with a higher degree of automation make long-term and highly time-resolved INP measurement more easily available (Bi et al., 2019; Brunner and Kanji, 2021; Möhler et al., 2021)"

Sect. 2.1 & P14L330-334: The authors use the meteorological data measured at the airport and connect these to their aerosol measurements and sampling activities carried out at the mountain station, which seems several kilometers apart at least. The authors might want to clarify that the meteorological data were not available at the station in the text. Also, please include a justification and discussion of why they think it is reasonable to refer to the met data at the airport for their studied aerosol properties from the mountain station in the manuscript.

We performed wind direction and speed measurements at the Cerro Mirador station, but the measurement only started at end of September 2019. The figure below shows the time series of wind direction and speed at Cerro Mirador station (in red) and the airport (in blue). Wind direction and speed are comparable at both stations. When doing the comparison of meteorology parameters, we always show the probability density function over 7-14 days.

Moreover, we used the cumulative precipitation along the backward trajectory ($P_{BT}$). Considering the spatial resolution (1-degree latitude-longitude) of the HYSPLIT model, $P_{BT}$ should not show a difference between the two stations. Temperature and pressure are considered to be uniform on a larger scale. Therefore, we assume that met data at the airport can be used for this study. At the end of Sect. 2.1, we included the following:

"These data were used as there was no meteorological data available at Cerro Mirador at the time of sampling. However, based on more recently installed meteorological measurements at Cerro Mirador we know, that wind speed as well as wind direction there are similar to those at the airport."

[Figure]

P3L74-76: Please address if any dryers were installed upstream of MPSS and APS. If the authors conducted the 'wet' particle size distribution measurements without a dryer, please address why dry particle diameter measurements, which are typically used for the n_s estimation, were not carried out and associated limitations (if any).

Indeed, there was a Nafion dryer installed in front of these two in-situ particle size measurements which we explicitly mention now in the text.

P3L76: The reviewer wonders why the 800 nm pore size filter was chosen to be used for this study (instead of the one with a smaller pore size). By looking at the particle size distribution figures in Sect. 3, it appears that a substantial fraction of <800 nm particles were measured during the DACAPO-PESO campaign. Later, the authors discuss the key feature of large particles (>500 nm). Therefore, there seemingly is a logical size leap between 500-800 nm. Please elaborate on the discussion/limitation.

The choice of the pore size was based on a balance between resistance and collection efficiency. It is important to note that 800 nm pore size does not mean that particles smaller than 800 nm are not collected. There is a size-dependent collection efficiency of different pore sizes. We based our choice on the study by Soo et al (2016), who, for the here used type of filter and flow, give a collection efficiency of 96%. Moreover, we compared results for the use of different filter pore sizes (200 nm and 800 nm) on different occasions and got the same results wrt. INP concentrations. This was presented e.g., at EAC 2020 (Abstract title: "Variations in off-line filter sampling and analysis for Ice Nucleating Particle measurements") and will be included in a publication which is currently in preparation.

P3L85-88: The authors might consider explaining the purpose of keeping samples frozen. What was the typical time span between sampling and offline INP measurements? All INP measurements were completed within a certain amount of time after the sample was taken? The aerosol sampling was done for 1-2 weeks (P3L77), and the filter was exposed to the ambient air and temperature during sampling while the filter is kept unfrozen. Do the authors think such a long sampling interval impacts the degradation of collected aerosol particles over the employed sampling time? Please include the discussion in the manuscript.

Our past (unpublished) experience showed that samples degraded over the course of a couple of weeks when they were not stored frozen (both, at room temperature and also in a refrigerator at +4°C). Related issues were described by a publication by Beall et al. (2020), who recommend storage at -20°C as the method causing the least changes in INP concentrations. We mention this in our text now.

Admittedly, sampling degradation may occur while the filter is being sampled. This, as also Beall et al. (2020) state, may cause artifacts. However, we see rather unexpected high concentrations, so the main tendencies reported in our study are very likely not affected.

P4L91: How did the authors determine the suspending water amount to be 3-4 mL? Based on the set lower detection limit of detectable INPs in the authors' offline assay? Please include the clarification in the manuscript.

First, it is important to mention that the lower the amount of water, the higher the INP concentration in the suspension. High concentrations in the suspensions are better for the INP measurements due to the generally low airborne INP concentrations, in combination with the instrument detection limit. Second, for LINA and INDA measurements and the additional heating test, a minimum amount of water is needed. When starting with our analysis for this study, we considered the INP concentration may be pretty low at Punta Arenas. For the first few test samples, we used the minimum amount of washing water, i.e., 3 mL. However, all of the samples are used up if only 3 mL are used. After testing a few samples, we realized the INP number concentration is high enough to use 4 mL of washing water. From then on, we used 4 mL of water, as this makes handling during the measurements much easier.

P15L346-361: The authors might consider including the specs of the back trajectory analysis method somewhere in the manuscript (even in the method section). It seems that the back trajectory analysis is one of the crucial analytical tools in this study. The reviewer wonders if the authors can incorporate the air mass height information in Figs. 8 and 11. Do the back trajectory data support that air masses travel through the free tropospheric conditions?

We feel that the trajectory analysis does not need an extra section, but added the following in Line 329:

"Backward trajectory analyses were performed with the HYSPLIT (HYbrid Single-Particle Lagrangian Integrated Trajectory) model (Stein et al., 2015), based on GDAS (Global Data Assimilation System) meteorological data."

We also did an analysis as suggested by you, deriving the frequencies that air mass spent at different altitudes during their passage to our measurement station. These are now included in the manuscript, amending the earlier Figures 8 and 11.

We added the following in L361: "This difference in air mass origin can also be seen in the right panels of Fig. 8, clearly showing that collected air masses spent a higher fraction of time in the free troposphere for sample 11."

[Figure]

Figure 8. Relative frequency of backward trajectories ending at Cerro Mirador site (600 m a.s.l.) with 1 h resolution, based on a 1° by 1° grid size (left) and the corresponding relative frequency of backward trajectory height (right) during sample 11 (a) and sample 12 (b).

[Figure]

Figure 11. Relative frequency of backward trajectories ending at Cerro Mirador site (600 m a.s.l.) with 1 h resolution, based on a 1° by 1° grid size (left) and the corresponding relative frequency of backward trajectory height (right) during cold season high (CH, panel a), warm season high (WH, panel b), and warm season low (WL, panel c).

P15L369-370: "A general enhancement of aerosol concentrations …" This sounds ambitious and speculative. So do the authors are suggesting that the aerosol intrusion from BL bumps up the aerosol concentration? Please detail what's meant by this sentence in the manuscript.

The measurement station is at 622 m a.s.l., and it is still inside the boundary layer. We did not mean the aerosol intrusion from BL.

Here we mean the aerosol concentration in the boundary during sample 12 is higher than that during sample 11, which is indicated by the vertical profile of the particle backscatter coefficient.

It is clearly explained in Lines 364-365:

"Also, the particle backscatter coefficient is higher throughout the boundary layer depth and above, indicating a generally higher aerosol load observed for sample 12."

But to prevent misunderstandings, we replaced "A general enhancement of aerosol concentrations is evident …" with "Generally, higher aerosol concentrations are evident …"

P21L433-435 & P19L403-405: These sentences are provoking but sound very ambitious. While this reviewer agrees that the ground-level INP & aerosol measurements are important, verifying what's stated in these parts (especially on P21) seemingly requires more measurements and evidence to constrain vilifications (e.g., cloud height INP data, vertical distribution of INPs over the studied site, online biological particle concentration measurements, INP measurement in precipitation samples etc.). Perhaps, the authors soften the tone of this sentence and also include the list of limitations and things to do as a future outlook? This reviewer thinks the limitations/outlooks are as important as good scientific results.

We exchanged the word "can" with "may" in the sentence, and added, following this sentence: "However, more research, combining in-situ and offline INP sampling with remote sensing methods is needed before this claim can fully be made."

P22L472-475: A chemical analysis of the aerosol particles would definitely add some complementary information (at least for the aerosol source identification), but the aerosol composition is not necesarrily the same as the INP composition. How about biological particle composition & rainwater INP & vertical distribution & ice crystal residuals measurement?

We inserted the following:

"Additionally, a combination with measurements of biological particles may be helpful, or examinations of INP in rainwater."

[Technical comments]

P2L36: INPs – already abbreviated previously.

Changed.

Figs. 8 & 11 captions: 600 m AGL or AMSL?

It is 600 MSL. Kmsl = 1 when doing HYSPLIT calculation.

P19L402-403: "of all discussed…" The reviewer does not understand this sentence. Please rephrase it.

We changed it to: " Among above discussed…"

[Misc. feedback]

Sects. 3.2 & 3.4: The reviewer finds the method used to estimate the biogenic INPs contribution (i.e., F_bio-INP) interesting and useful yet speculative. In future fieldwork in a similar location, the deployment of an online biological particle sensor and linking such online data to F_bio-INP can even add/solidify more value to this F_bio-INP idea, in the reviewer's opinion.

Thanks for this comment and hint.

**References**

Schrod, J., Thomson, E. S., Weber, D., Kossmann, J., Pöhlker, C., Saturno, J., Ditas, F., Artaxo, P., Clouard, V., Saurel, J.-M., Ebert, M., Curtius, J., and Bingemer, H. G.: Long-term deposition and condensation ice-nucleating particle measurements from four stations across the globe, Atmospheric Chemistry and Physics, 20, 15 983–16 006, https://doi.org/10.5194/acp-20-15983-2020, 2020.

Schneider, J., Höhler, K., Heikkilä, P., Keskinen, J., Bertozzi, B., Bogert, P., Schorr, T., Umo, N. S., Vogel, F., Brasseur, Z., Wu, Y., Hakala, S., Duplissy, J., Moisseev, D., Kulmala, M., Adams, M. P., Murray, B. J., Korhonen, K., Hao, L., Thomson, E. S., Castarède, D., Leisner, T., Petäjä, T., and Möhler, O.: The seasonal cycle of ice-nucleating particles linked to the abundance of biogenic aerosol in boreal forests, Atmospheric Chemistry and Physics, 21, 3899–3918, https://doi.org/10.5194/acp-21-3899-2021, 2021.

Testa, B., Hill, T. C. J., Marsden, N. A., Barry, K. R., Hume, C. C., Bian, Q., Uetake, J., Hare, H., Perkins, R. J., Möhler, O., Kreidenweis, S. M., and DeMott, P. J.: Ice Nucleating Particle Connections to Regional Argentinian Land Surface Emissions and Weather During the Cloud, Aerosol, and Complex Terrain Interactions Experiment, Journal of Geophysical Research: Atmospheres, 126, e2021JD035 186, https://doi.org/https://doi.org/10.1029/2021JD035186, e2021JD035186 2021JD035186, 2021.

Bi, K., McMeeking, G. R., Ding, D. P., Levin, E. J. T., DeMott, P. J., Zhao, D. L., Wang, F., Liu, Q., Tian, P., Ma, X. C., Chen, Y. B., Huang, M. Y., Zhang, H. L., Gordon, T. D., and Chen, P.: Measurements of Ice Nucleating Particles in Beijing, China, Journal of Geophysical Research: Atmospheres, 124, 8065–8075, https://doi.org/https://doi.org/10.1029/2019JD030609, 2019.

Brunner, C. and Kanji, Z. A.: Continuous online monitoring of ice-nucleating particles: development of the automated Horizontal Ice Nucleation Chamber (HINC-Auto), Atmospheric Measurement Techniques, 14, 269–293, https://doi.org/10.5194/amt-14-269-2021, 2021.

Möhler, O., Adams, M., Lacher, L., Vogel, F., Nadolny, J., Ullrich, R., Boffo, C., Pfeuffer, T., Hobl, A., Weiß, M., Vepuri, H. S. K., Hiranuma, N., and Murray, B. J.: The Portable Ice Nucleation Experiment (PINE): a new online instrument for laboratory studies and automated long-term field observations of ice-nucleating particles, Atmospheric Measurement Techniques, 14, 1143–1166, https://doi.org/10.5194/amt-14-1143-2021, 2021.

Soo, J. C., K. Monaghan, T. Lee, M. Kashon, and M. Harper (2016), Air sampling filtration media: Collection efficiency for respirable size-selective sampling, Aerosol Sci. Technol., 50(1), 76-87, doi:10.1080/02786826.2015.1128525.

Beall, C. M., D. Lucero, T. C. Hill, P. J. DeMott, M. D. Stokes, and K. A. Prather (2020), Best practices for precipitation sample storage for offline studies of ice nucleation in marine and coastal environments, Atmos. Meas. Tech., 13, 6473-6486, doi:10.5194/amt-13-6473-2020.

Stein, A. F., Draxler, R. R., Rolph, G. D., Stunder, B. J. B., Cohen, M. D., and Ngan, F.: NOAA's HYSPLIT Atmospheric Transport and Dispersion Modeling System, Bulletin of the American Meteorological Society, 96, 2059–2077, https://doi.org/10.1175/bams-d-14-00110.1, 2015.

---

## Author Comment (AC2)

Dear Reviewer,

We thank you for doing this review and for your suggestions that helped to improve our manuscript. Below, please find your original comments in blue and our responses in black. When referencing page and line numbers, we are always referring to the old version of the manuscript.

**General Comment**

In the current manuscript, aerosol particle samples were collected close to Punta Arenas, Chile, on Cerro Mirador at the southernmost tip of South America between May 2019 and March 2020. The objective was to evaluate the capability of these particles to act as ice nucleating particles (INP) and to identify their possible sources. Also, the INP data were complemented with meteorological data, air mass backward trajectories, and lidar observations. I consider that this study is relevant for the ice nucleation community and it is especially important for the Southern Hemisphere due to the lack of studies in this region. Also, I think that the database generated in this study is valuable because of the long-term sampling time. The present manuscript is within the ACP scope and it can be accepted for its publication once the following minor comments are considered.

**Minor comments:**

**Line 6** Why do the authors state that 90% and 80% of the INP are proteinaceous material. I mean, how did you calculate these percentages? I could not find any chemical analysis to determine the quantity of proteinaceous material within the samples.

It is explained in P8 Line 255 that protein-based biogenic INPs will lose their ice nucleating ability when heated at 95 °C for 1 hour, so that the remaining INPs were labeled as heat-resistant INPs. By using equation (6), i.e., by relating the original INP concentration with that obtained after heating, we can calculate the proteinaceous-based INP number fraction.

**Line 58** In this part the authors mention that it is important to perform INP measurements in the Southern Ocean; however, the samples were collected in a mountain close to the coast and not in

the open ocean. Therefore, it is very likely that the present results were influenced by continental aerosol sources.

How far away is Punta Arenas from Cerro Mirador? Can anthropogenic particles emitted in Punta Arenas influence the measurements performed in Cerro Mirador? Also, are there fertile soils close to your sampling site? Or why did you compare your results with O'Sullivan et al. (2014)? The description of the sampling site needs to be improved.

The Cerro Mirador measurement station is southwest of the Punta Arenas city, about 11.6 km away. We modified the description of the measurement site and added the following figure in the supplement to show the location of three measurement stations. We also added, to the sentence you refer to here: "and in the Southern Hemisphere in general."

[Figure]

The anthropogenic particles are not expected to influence the INP measurement for the following two reasons. (1) The backward trajectories show that the air parcel mainly came from the west to the Cerro Mirador station. The land cover on the west of Cerro Mirador is mainly forest and herbaceous vegetation, and the open ocean. (2) Anthropogenic pollution is not a major source of INP at temperatures above -20 °C, even in a very polluted megacity (Chen et al., 2018).

We modified the measurement site description and added one paragraph at the beginning of Sect. 2.1 Measurement sites to introduce the landscape of Punta Arenas and Cerro Miradors.

"Punta Arenas is one of the largest cities in the Patagonia region, with a population of more than 100000. It is about 1419 km from the coast of Antarctica. To the west of Punta Arenas is a

mountain area and the landscape is mainly forest and herbaceous vegetation, whereas to the east of Punta Arenas there is relatively flat terrain, with a landscape of herbaceous vegetation."

Considering the geographical feature of Punta Arenas and air parcel origins, the conventional thinking previous to our study was, that INPs at Punta Arenas are from the marine aerosol in the Southern Ocean. Nevertheless, Punta Arenas is indeed surrounded by at least 150 km of mountainous land interspersed with fjords in all directions, and we clearly saw higher INP concentrations than reported for the Southern Ocean, biogenic in origin, likely originating from terrestrial sources. This is the main reason why we compared our results with O'Sullivan et al. (2014) and Tobo et al. (2013). Our results also point out that already the tip of South America can be an important land source of INP in the Southern Ocean, at least in the vicinity of the continent.

**Line 76** I suggest changing the word "aerosols" to aerosol particles. Also, it is mentioned that the used filters contain a pore size of 800 nm; however, afterward it is mentioned that the analysis was performed for particles >500 nm. Please explain this discrepancy.

We exchanged it to "aerosol particles".

As for the pore size, it is important to note that 800 nm pore size does not mean that particles smaller than 800 nm are not collected. There is a size-dependent collection efficiency of different pore sizes. We based our choice on the study by Soo et al (2016), who, for the here used type of filter and flow, give a collection efficiency of 96%. Moreover, we compared results for the use of different filter pore sizes (200 nm and 800 nm) on different occasions and got the same results wrt. INP concentrations. This was presented e.g., at EAC 2020 (Abstract title: "Variations in off-line filter sampling and analysis for Ice Nucleating Particle measurements") and will be included in a publication which is currently in preparation.

So it can be assumed that we collected particles down to much smaller sizes than 800 nm. However, INP may still mostly be larger particles (as e.g. found in Mason et al., 2016, Creamean et al., 2018, and Gong et al., 2020). That is at the core of using particle number concentration for particles > 500 nm for INP parameterizations, as originally suggested by DeMott et al. (2010)

(cited in our text). Therefore, for testing this parameterization by DeMott et al. (2010), we needed to use particle number concentration for particles > 500 nm as an input parameter. Therefore, it is also useful to compare particle number concentration for particles > 500 nm as derived from our in-situ ground-based measurement with those from lidar measurements, which happen to have the means to derive particle number concentrations for this size range.

**Line 78** What was the used aerosol sampler?.

The sampler used in this study is a 47 mm Single Stage Filter Assembly from Savillex (https://www.savillex.com/en/product/filtration-assemblies/47-mm-single-stage-filter-assembly-14-x-14-tefzel-clamp--401-21-47-30-21-2?pageid=19).

**Line 85** Can the authors please elaborate more on the sample storage protocol? Is there any chance that the samples could have experienced a memory effect when stored at -20C? For how long were the samples stored prior to the ice nucleation analysis? Did the authors evaluate the impact of long-term storage?

A memory effect due to frozen storage to our knowledge has not been reported in the literature, nor been observed by any colleague doing these kinds of studies. Instead, our past (unpublished) experience (and that of others) showed that samples degraded over the course of a couple of weeks when they were not stored frozen (both, at room temperature and also in a refrigerator at +4°C). Related issues were described by a publication by Beall et al. (2020), who recommend storage at -20°C as the method causing the least changes in INP concentrations. We mention this in our text now.

Concerning the lengths of storage, the samples were collected from May 2019 to March 2020. The samples were measured from 26 May 2020 to 18 August 2020 at TROPOS, Germany. Once the samples were taken out of the freezer, the filter sample washing and INP measurement is completed within 2 hours. Based again on Beall et al. (2020), we do generally not expect an effect from our storage on our samples.

**Line 94-95** I am wondering if the PCR-tray with the sample was heated or if the heat was only applied to the sample?

We pipetted the samples into the PCR tray, sealed the PCR tray with a foil, and then heated the PCR tray together with the samples.

**Line 121** I believe that "V" corresponds to the volume of the liquid deposited on each well instead of the air volume. Please clarify this.

If the "V" represents the volume of the liquid deposited on each well, then the calculated $N_{INP}$ is the INP number concentration in the water samples. That can be used when samples such as precipitation or sea water is examined, to obtain concentration per liter of water.

Here we use the "V" as the air volume which, literally speaking, was collected into each well, so that the calculated $N_{INP}$ is the INP number concentration in the ambient air.

**Line 149 - 150** Describe in more detail how ns was calculated.

We think that equation (5) together with the text here provides the full description of what was done.

**Line 175-178** This classification should be in the methods section.

The classification does not belong to the method section. The idea of the classification only starts to make sense after one has seen the measured spectra of $N_{INP}$ in Fig. 1. Mentioning this part of the further evaluation already in the method section will disturb the logical flow. So we think it is better to keep it as it is.

**Line 203** I do not understand why the authors compare their results with those from Tobo et al. (2013) and the former study was performed on aerosol particles from a forest while the present study focuses on coastal particles.

Please see our reply for the 2nd minor comment.

**Line 228-229** Why did you choose these temperatures or do they refer to a range of temperatures?

These temperatures were chosen, as they cover a large temperature range and as for each of them, a considerable amount of data points exist.

We did not choose temperatures below -18 or above -8 °C because, for a large number of samples, all or no examined droplets were frozen, i.e., the related INP concentrations are outside of the detection range of our instruments, so that no data exists.

**Figure 3b** I did not completely get the purpose and meaning of panel b.

The boxplot in Fig. 3b shows which fraction of INP originated from biogenic particles at different temperatures (based on results from equation 6). The red line, corresponding to the right axis, shows how many samples contribute to this statistical analysis. As mentioned in our answer to your previous remark, at the very low and high temperatures, the INP concentrations of some samples are out of instrument detection limits. Therefore, the red line shows low values at very low and high temperatures.

In panel b, we show that (1) most of the INPs are biogenic particles in origin, and (2) the fraction of biological INPs is getting lower at lower temperatures.

We think we explain this clearly in the text L241-246. We however added at the end of the figure caption: "(relating to the right axis)".

**Line 269** According to Figure 4, the black dots correspond to the heated samples and the red dots to the unheated samples; however, in line 269 the authors state the opposite. Also, this is not consistent between the text in the figure caption and the figure legend.

We corrected the color in Fig.4.

 Why Samples 11 and 12 were selected for the case study? It is unclear to me what was the motivation to have the Case study and also the cluster analysis performed in Section 3.5. Is it not better to remove the Case study and to add those 2 samples to the cluster analysis shown in Section 3.5?

Sample 11 is the only sample that shows low $N_{INP}$ during the cold season, this is why choose sample 11 for the case study. Sample 12 is just sampled close to 11, so while we expected similar conditions, the $N_{INP}$ of Sample 12 is much higher. We have explained this in Lines 324-326.

Examining these two samples in detail is highly valuable, as sample 11 is very special. Adding this to the cluster analysis would take away the here gained information, and we think it is worth going into such detail on the comparison between these subsequent Samples 11 and 12.

 Please add the used software to get the air mass the backward trajectories in the Methods.

We added the following in P14 L329:

"Backward trajectory analyses were performed with the HYSPLIT (HYbrid Single-Particle Lagrangian Integrated Trajectory) model (Stein et al., 2015), based on GDAS (Global Data Assimilation System) meteorological data."

 I am wondering how valid it is to correlate the meteorological data from the airport with the INP data from the Cerro Mirador.

We performed wind direction and speed measurements at the Cerro Mirador station, but the measurement only started at end of September 2019. The figure below shows the time series of wind direction and speed at Cerro Mirador station (in red) and the airport (in blue). Wind direction and speed are comparable at both stations. When doing the comparison of meteorology parameters, we always show the probability density function over 7-14 days.

Moreover, we used the cumulative precipitation along the backward trajectory ($P_{BT}$). Considering the spatial resolution (1-degree latitude-longitude) of the HYSPLIT model, $P_{BT}$ should not show a difference between the two stations. Temperature and pressure are considered

to be uniform on a larger scale. Therefore, we assume that meteorology data at the airport can be used for this study. At the end of Sect. 2.1, we included the following:

"These data were used as there was no meteorological data available at Cerro Mirador at the time of sampling. However, based on more recently installed meteorological measurements at Cerro Mirador we know, that wind speed as well as wind direction there are similar to those at the airport."

[Figure]

**Line 401** Based on the heat treatment analysis the authors state that some INPs were of biogenic origin; however, it is unclear what is the source of bulk INPs, i.e., those that are not of biogenic origin?

This is a good question. We assume the heat-resistance INPs in this study are mainly from soil dust, with little contribution from the sea-spray aerosol. Unfortunately, we can not test the particle composition after the heating process. As we can not give any more information on these

particles, we did not change anything in the text. If you prefer us to muse about this in the text, please let us know, but we would prefer to not do this.

**References**

Chen, J., Wu, Z., Augustin-Bauditz, S., Grawe, S., Hartmann, M., Pei, X., Liu, Z., Ji, D., and Wex, H.: Ice-nucleating particle concentrations unaffected by urban air pollution in Beijing, China, Atmos. Chem. Phys., 18, 3523–3539, https://doi.org/10.5194/acp-18-3523-2018, 2018.

Soo, J. C., K. Monaghan, T. Lee, M. Kashon, and M. Harper (2016), Air sampling filtration media: Collection efficiency for respirable size-selective sampling, Aerosol Sci. Technol., 50(1), 76-87, doi:10.1080/02786826.2015.1128525.

Mason, R. H., M. Si, C. Chou, V. E. Irish, R. Dickie, P. Elizondo, R. Wong, M. Brintnell, M. Elsasser, W. M. Lassar, K. M. Pierce, W. R. Leaitch, A. M. MacDonald, A. Platt, D. Toom-Sauntry, R. Sarda-Esteve, C. L. Schiller, K. J. Suski, T. C. J. Hill, J. P. D. Abbatt, J. A. Huffman, P. J. DeMott, and A. K. Bertram (2016), Size-resolved measurements of ice-nucleating particles at six locations in North America and one in Europe, Atmos. Chem. Phys., 16(3), 1637-1651, doi:10.5194/acp-16-1637-2016.

Creamean, J. M., R. M. Kirpes, K. A. Pratt, N. J. Spada, M. Maahn, G. de Boer, R. C. Schnell, and S. China (2018), Marine and terrestrial influences on ice nucleating particles during continuous springtime measurements in an Arctic oilfield location, Atmos. Chem. Phys., 18, 18023–18042, doi:10.5194/acp-18-18023-2018.

Gong, X., H. Wex, M. van Pinxteren, N. Triesch, K. W. Fomba, J. Lubitz, C. Stolle, B. Robinson, T. Müller, H. Herrmann, and F. Stratmann (2020), Characterization of aerosol particles at Cape Verde close to sea and cloud level heights - Part 2: ice nucleating particles in air, cloud and seawater, Atmos. Chem. Phys., 20, 1451-1468, doi:10.5194/acp-20-1451-2020.

Beall, C. M., D. Lucero, T. C. Hill, P. J. DeMott, M. D. Stokes, and K. A. Prather (2020), Best practices for precipitation sample storage for offline studies of ice nucleation in marine and coastal environments, Atmos. Meas. Tech., 13, 6473-6486, doi:10.5194/amt-13-6473-2020.

Stein, A. F., Draxler, R. R., Rolph, G. D., Stunder, B. J. B., Cohen, M. D., and Ngan, F.: NOAA's HYSPLIT Atmospheric Transport and Dispersion Modeling System, Bulletin of the American Meteorological Society, 96, 2059–2077, https://doi.org/10.1175/bams-d-14-00110.1, 2015.

---

## Author Response (AR2)

**Response to Editor's comment**

Please find your original comments in blue and our responses in black. When referencing page and line numbers, we are always referring to the latest version of the manuscript.

**Comments to the author**:

I would like to thank the authors for incorporating the suggestions made by both reviewers. The revised version looks pretty good and it is almost ready for publication; however, I have the following comments before I can accept the manuscript.

Minor Comment:

Please state somewhere that the relevant heterogeneous ice nucleation mode in the present study is immersion freezing, briefly indicating why this mode is important for mixed-phase clouds.

We added the following in Line 23:

"Heterogeneous ice nucleation processes can occur via different pathways, e.g., immersion freezing, deposition nucleation, condensation freezing, and contact freezing (Pruppacher and Klett, 1998; Hoose and Möhler, 2012). Among these, immersion freezing (ice nucleation by solid particles immersed in super-cooled water) is the dominant process for ice nucleation in mixed-phase clouds (Murray et al., 2012)."

We modified the following in Line 76 to clarify that we focus on immersion freezing.

"Secondly, we focus on the immersion freezing behavior of aerosol particles and the derived INP spectra are presented."

Technical Comments:

Line 6: Based on the heating test, roughly….

We change it to: "Heating of the samples revealed that roughly".

Line 8: with AN agricultural area in Argentina

Done.

Lines 23: Add a reference after "heterogeneous ice nucleation".

Done.

Line 24: I think "the frozen ice crystals" is not completely appropriate. How about "nucleated ice crystals"?

We think it is better to remove the "frozen" here, as the "nucleation" is the "birth" of the ice crystal.

Line 34: Add a reference after "Southern Hemisphere".

Done.

Line 44: "than previously found and found that these". Please rewrite this.

Changed to "than previous studies and found that these"

Line 51: "the real world". I think this term is not fully appropriate.

Change it to "Earth". Now is:

"Since Earth has large spatial heterogeneity and temporal variability,…"

Line 65: Please fix the following sentence "particular, and this is true worldwide and, due".

Change it to:

"Long-term measurements with high time resolution are needed to understand the aerosol-cloud interaction in general and INP in particular. This is true worldwide and particularly in the region of the Southern Ocean and Southern Hemisphere, due to a still existing lack of data."

Line 77: Please add the word "inhabitants" after "more than 100000" and also add the corresponding reference.

We added "inhabitants". The population information is from Wikipedia. Not sure if it is proper to cite Wikipedia information.

Line 79: Add a reference after "landscape of herbaceous vegetation".

Done.

Line 90: Please add the details of the used sampler.

If I understand correctly, you are referring to Line 89, the sampler for the INP filter.

We added the following:

"The sampler used in this study is a 47 mm single-stage filter holder (Savillex LLC, MN, USA)."

Line 95: Replace "based on more recently" by "based on a recently"

Done. We removed "more", but did not add "a" because the following word "measurements" is plural.

Line 101: Please state for how long the samples were stored under this conditions previous to their INP analysis.

We added the following:

"The measurements were performed from May to August 2020, and the samples were stored in a freezer for roughly 3-8 months before they are analyzed."

Line 163: Given that an INP sample was collected for several days, please indicate if the used A to calculate ns corresponds to the average value for the 7-14 day periods.

We modified and now it is:

"where A is the averaged particle surface area concentration during the corresponding INP collection periods."

Figure 2: I am wondering if this figure shows circles or dots. Either make the circles bigger or change "circles" by "dots", as well, the figure legend.

Changed "circles" to "dots".

Line 381: Please avoid using twice "in summary" in the same paragraph.

We removed the 2nd "in summary".

Line 456: "Punta Arenas (Chile)."

Done.

Line 456: Delete PSSD.

Done.

Figure S1: Replace "surrounding the Punta Arenas" with "surrounding Punta Arenas". Delete the "References" subsection in the SI (page 10).

Done.